# *Q-Palette*: Fractional-Bit Quantizers Toward Optimal Bit Allocation for Efficient LLM Deployment

**Deokjae Lee**[1,2]      **Hyun Oh Song**[1,2*]

[1]Seoul National University, [2]Neural Processing Research Center

{bdbj, hyunoh}@mllab.snu.ac.kr

## Abstract

We study weight-only post-training quantization (PTQ), which quantizes the weights of a large language model (LLM) without retraining, using little or no calibration data. Weight-only PTQ is crucial for reducing the memory footprint and latency of LLM inference, especially in memory-bound, small-batch inference scenarios, such as personalized inference on edge devices. Despite its importance, irregular weight distributions with heavy-tailed outliers in LLMs complicate quantization, recently motivating rotation-based methods that transform weights into near-Gaussian distributions, which are more regular with fewer outliers, thereby reducing quantization error. In this work, we first derive the information-theoretically optimal bit allocation for Gaussianized weights under given bit budgets, revealing that fine-grained fractional-bit quantizers approaching the Gaussian distortion-rate bound are essential to achieve near-optimal quantization performance. To bridge this theoretical insight and practical implementation, we introduce *Q-Palette*, a versatile collection of fractional-bit quantizers that range from trellis-coded quantizers offering near-optimal distortion to simpler vector and scalar quantizers optimized for faster inference, all efficiently implemented with optimized CUDA kernels across various bitwidths. Furthermore, leveraging Q-Palette as a foundational component, we propose a novel mixed-scheme quantization framework, jointly optimizing quantizer choices and layer fusion decisions given resource constraints. The code is available at https://github.com/snu-mllab/Q-Palette.

## 1   Introduction

Large language models (LLMs) have recently achieved significant success across diverse tasks and are increasingly being deployed on resource-limited edge devices, such as laptops or smartphones [39, 9, 33]. However, these edge devices typically have limited memory resources and often process small-batch workloads, making inference severely memory-bound. Weight-only quantization has thus become essential, allowing models to achieve significantly greater compression at similar levels of performance compared to quantizing both weights and activations. Moreover, recent studies have demonstrated that weight-only quantization, beyond its well-known compression advantages, can also significantly accelerate inference speed in small-batch decoding scenarios by alleviating memory bottlenecks [28, 40, 20]. Specifically, we address weight-only post-training quantization (PTQ), enabling model quantization without costly retraining or extensive calibration data, which are common constraints in real-world deployments [5, 24].

However, quantizing LLM weights remains challenging due to inherently irregular, heavy-tailed distributions containing outliers that significantly broaden quantization ranges [23, 28, 29, 54]. To address this, recent research introduced a theoretically grounded approach known as *incoherence*

---

*Corresponding author

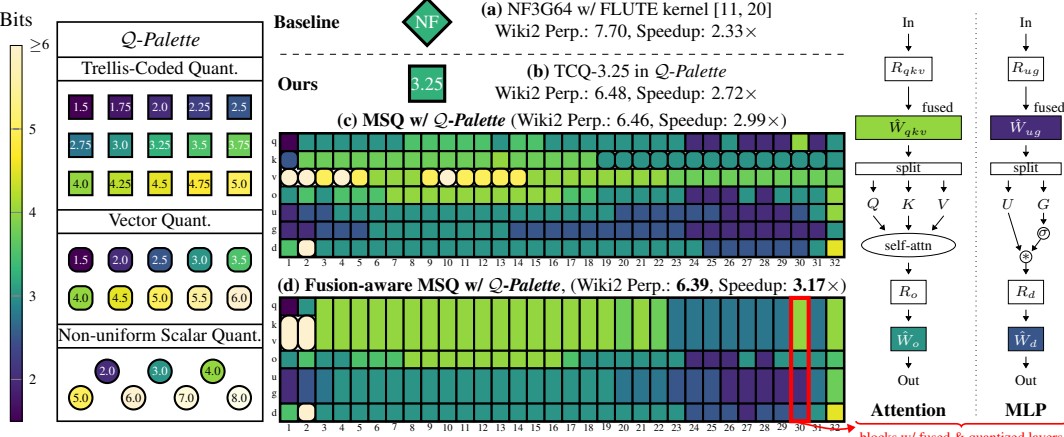

Figure 1: Qualitative comparison of quantization frameworks based on Q-Palette against the NormalFloat baseline with FLUTE kernels [11, 20], evaluated on the LLaMA 3.1-8B model using an RTX4090 GPU with a batch size of 1. Compared to (a) NormalFloat, (b) single-scheme quantization with TCQ-3.25 achieves a 17% inference speedup, (c) MSQ with Q-Palette provides a 28% speedup, and (d) fusion-aware MSQ further yields a 36% speedup alongside reduced WikiText2 perplexity, highlighting the practical effectiveness of Q-Palette and our MSQ framework. In the MSQ visualizations, columns represent transformer blocks, and rows represent linear layers, with colors indicating selected quantization bitwidths. The right visualization illustrates fused layers within the 30-th transformer block from configuration (d). Refer to Appendix F for the experimental details.

*processing*, which applies rotation matrices (*e.g.*, random Hadamard transforms) to weight matrices, reducing outliers by modifying distributions into approximately Gaussian forms [21, 6, 1, 49].

We begin with a natural question: if ideal Gaussian quantizers were available at arbitrary fractional bitwidths, how should we allocate bits across layers to minimize performance degradation under a fixed memory budget? Building upon the *linearity theorem* [35], which approximates performance degradation by quantization as a weighted sum of layer-wise mean squared errors, we derive an information-theoretically optimal bit allocation strategy for Gaussianized weights. Our analysis reveals that fine-grained fractional-bit quantizers that closely match their theoretical distortion bounds are essential to approaching the quantization performance predicted by theory. However, existing sophisticated Gaussian quantizers, such as trellis-coded quantization, have only been implemented with fused kernels for a limited set of integer bitwidths (*e.g.*, 2, 3, 4 bits), with little or no support for batch sizes larger than one [19, 15, 49, 50].

To bridge theoretical insights with practical quantization, we introduce *Q-Palette*, a versatile set of fractional-bit quantizers, ranging from trellis-coded quantizers (TCQ) for near-optimal distortion to simpler vector and scalar quantizers for low latency, covering diverse accuracy-latency trade-offs. We provide optimized CUDA kernels supporting a wide range of fractional bitwidths with broader batch size support than prior sophisticated quantization methods (*e.g.*, QTIP [50]). To enable even finer bitwidth control, we further propose half-TCQ, a novel TCQ variant that mixes two TCQ quantizers of different bitwidths within a single layer (*e.g.*, 2.5 and 3.0 bits) to realize intermediate bitwidths such as 2.75 bits, and extend our CUDA kernels to support this variant.

We integrate Q-Palette into a resource-constrained mixed-scheme quantization (MSQ) framework to demonstrate its practical utility. To further improve accuracy-latency trade-offs, we propose *fusion-aware MSQ*, the first MSQ approach that jointly optimizes quantizer selection and layer fusion, introducing a new optimization dimension (see Figure 1). Here, linear layers sharing the same input (*e.g.*, query, key, and value projections in a Transformer block) can be fused into a single linear layer, reducing memory accesses and kernel launches [26, 12, 40]. By incorporating layer fusion, fusion-aware MSQ achieves significant gains in accuracy-latency trade-offs. Extensive experiments on LLaMA 2, LLaMA 3, and Qwen models demonstrate that our MSQ framework with Q-Palette consistently outperforms strong data-free and data-aware weight-only PTQ baselines under both memory- and latency-constrained settings.

## 2 Preliminaries

### 2.1 Linearized surrogate objective for post-training quantization

Previous studies have approximated the performance degradation in neural networks induced by PTQ as a linear combination of layer-wise surrogate losses derived from second-order Hessian approximations [14, 7]. In the context of LLMs, where performance is typically measured by perplexity, the *linearity theorem* shows that the perplexity increase induced by quantization can be accurately approximated as a weighted sum of per-layer quantization errors [35]. This surrogate is especially valuable for data-free quantization scenarios where Hessian-based surrogate losses are unavailable. Formally, the linearized surrogate is expressed as:

$$\mathcal{L}(\{Q(W_l)\}_{l=1}^L) - \mathcal{L}(\{W_l\}_{l=1}^L) \approx \sum_{l=1}^L a_l \underbrace{\|Q(W_l) - W_l\|^2 / \|W_l\|^2}_{=: \, \mathrm{err}(Q;W_l)},$$

where $\mathcal{L}(\cdot)$ denotes the perplexity loss, $W_l \in \mathbb{R}^{d_l^{\mathrm{in}} \times d_l^{\mathrm{out}}}$ is the weight matrix of layer $l$, $Q(\cdot)$ is a quantization function, $a_l$ is the empirically estimated sensitivity coefficient for layer $l$, and $\mathrm{err}(Q; W_l)$ is the normalized quantization error of layer $l$.

Leveraging this surrogate, the memory-constrained MSQ problem with a set of candidate quantizers $\mathcal{Q}$ can be formulated as a multiple-choice knapsack problem (MCKP):

$$\underset{P_{lq} \in \{0,1\}}{\text{minimize}} \quad \sum_{l=1}^L a_l \left( \sum_{q=1}^{|\mathcal{Q}|} P_{lq} \cdot \mathrm{err}(Q_q; W_l) \right) \tag{1}$$

$$\text{subject to} \quad \sum_{q=1}^{|\mathcal{Q}|} P_{lq} = 1, \quad \forall 1 \le l \le L,$$

$$\sum_{l=1}^L \sum_{q=1}^{|\mathcal{Q}|} P_{lq} \cdot \mathrm{bit}(Q_q; W_l) d_l^{\mathrm{in}} d_l^{\mathrm{out}} \le M,$$

where $Q_q$ denotes a candidate quantizer, $\mathrm{bit}(Q_q; W_l)$ is the average number of bits per weight component for the weight matrix $W_l$ quantized by $Q_q$, $P_{lq} \in \{0, 1\}$ is a binary indicator selecting quantizer $Q_q$ for layer $l$, and $M$ denotes the total memory budget (in bits) allocated for quantized model [46]. This formulation explicitly casts MSQ as a combinatorial optimization problem grounded in a linearized performance surrogate, providing a principled framework for optimal bit allocation under strict memory constraints [35, 7].

## 3 Q-Palette: fractional-bit quantizers

### 3.1 Motivation and design goals

Building on the theoretical foundation introduced in Section 2, we now derive the information-theoretically optimal bit allocation strategy and discuss its implications for the design of practical quantizers. Under the assumption that weight matrices are Gaussianized via incoherence processing, the quantization problem can be viewed as a Gaussian source coding problem. In this setting, classical rate-distortion theory establishes a fundamental lower bound on expected quantization error as $\mathbb{E}[\mathrm{err}(Q)] \ge 2^{-2\mathrm{bit}(Q)}$ [10]. Assuming ideal Gaussian quantizers that achieve this bound at arbitrary fractional bitwidths $b_l \ge \eta$, the memory-constrained MSQ problem (1) simplifies to:

$$\underset{b_l \ge \eta}{\text{minimize}} \quad \sum_{l=1}^L a_l 2^{-2b_l} \tag{2}$$

$$\text{subject to} \quad \sum_{l=1}^L b_l d_l^{\mathrm{in}} d_l^{\mathrm{out}} \le M,$$

where $b_l$ is the fractional bitwidth allocated to layer $l$, and $\eta > 0$ is a minimum bitwidth threshold introduced to avoid degenerate cases such as assigning 0-bit to a layer. This formulation admits a closed-form solution as stated in Theorem 3.1.

Table 1: Quantizers, kernel implementations, and supported bitwidth intervals in Q-Palette.

| Quantization scheme | Kernel implementations | Supported bitwidths (bits) |
|---|---|---|
| Non-uniform scalar quantization (NUQ) | Tensor Core, CUDA Core | 2.0, 3.0, 4.0, 5.0, 6.0, 7.0, 8.0 |
| Vector quantization (VQ) | Tensor Core, CUDA Core | 1.5, 2.0, 2.5, 3.0, 3.5, 4.0, 4.5, 5.0, 5.5, 6.0 |
| Trellis-coded quantization (TCQ) | Tensor Core (TCQ) Tensor Core (Half-TCQ) | 1.5, 2.0, 2.5, 3.0, 3.5, 4.0, 4.5, 5.0 1.75, 2.25, 2.75, 3.25, 3.75, 4.25, 4.75 |

**Theorem 3.1** (Optimal bit allocation with ideal Gaussian quantizers). *If the budget $M$ is feasible, i.e., $M \geq \eta \sum_{l=1}^{L} d_l^{in} d_l^{out}$, then the optimal fractional bit allocation $\{b_l^*\}$ for problem (2) is given by*

$$b_l^* = \max \left\{ \eta, \frac{1}{2\ln(2)} \left( \ln \frac{a_l}{d_l^{\text{in}} d_l^{\text{out}}} \right) + C \right\}, \quad \forall 1 \leq l \leq L,$$

*for the constant $C$ that satisfies the memory constraint $\sum_l b_l^* d_l^{\text{in}} d_l^{\text{out}} = M$.*

*Proof.* See Appendix A. □

In practice, however, quantization must be performed using a finite set of non-ideal quantizers $\mathcal{Q} = \{Q_1, \ldots, Q_N\}$, which introduces a gap between theoretical optimality and actual achievable performance. The extent of this gap depends primarily on two factors: (1) how closely each quantizer approaches the ideal distortion bound $2^{-2\text{bit}(Q)}$, and (2) how finely the available bitwidths can approximate the optimal fractional bit allocations $b_l^*$. Please refer to Appendix B for further analysis.

This motivates the need for practical quantizers that are both accurate and available at fine-grained fractional-bit intervals. Moreover, quantizers often exhibit a trade-off between distortion and computational efficiency: more sophisticated quantizers may offer lower error but incur higher inference costs. These considerations motivate the design of a practical quantization suite that supports fine-grained fractional bitwidths while also accounting for trade-offs between quantization error and computational efficiency. Guided by these goals, we design Q-Palette as a versatile collection of quantizers tailored to balance quantization error and computational efficiency across deployment scenarios.

### 3.2 Quantization schemes in Q-Palette

Q-Palette includes three quantizer families, non-uniform scalar quantization (NUQ), vector quantization (VQ), and trellis-coded quantization (TCQ), spanning a range of quantization error and inference latency trade-offs. In this section, we briefly introduce these quantization schemes.

**Non-uniform scalar quantization.** NUQ quantizes each scalar weight by assigning it to an entry in a non-uniformly spaced lookup table (LUT), in contrast to uniform scalar quantizers, which use equally spaced intervals [27]. We construct NUQ codebooks via $k$-means clustering on random Gaussian samples [34], thus optimizing the codebook entries for Gaussianized weights. NUQ incurs low dequantization overhead and enables efficient inference [20, 40].

**Vector quantization.** VQ partitions weight vectors into groups of fixed dimension, assigning each group to the nearest vector entry in a precomputed codebook [51]. In Q-Palette, we specifically implement 2D VQ, generating codebooks via $k$-means clustering on random 2D Gaussian samples. Efficient kernel implementations rely on LUTs whose sizes are powers of two, enabling compact bit-level representations. Consequently, fractional bitwidths at $0.5$-bit intervals become natural can-

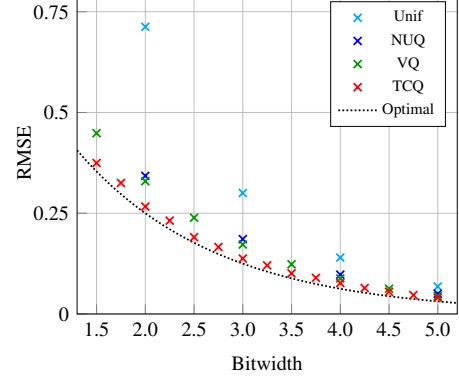

Figure 2: Gaussian quantization error of Q-Palette quantizers (NUQ, VQ, TCQ) compared to the uniform baseline.

didates for efficient implementations. For instance, a $(2^3, 2)$-shaped LUT (eight 2D vectors) encodes two elements with 3 bits, effectively achieving $1.5$ bits per scalar weight. Such constructions enable VQ to support fractional bitwidths at intervals of $0.5$ bits.

Table 2: Decoding-latency speedup of quantized LLaMA 3.1-8B models relative to the FP16 baseline on an RTX 4090 GPU. 'TC' and 'CC' denote Tensor Core and CUDA Core kernels, respectively.

| Decoding-latency speedup compared to FP16 (batch size = 1) | | | | | | | | | |
|---|---|---|---|---|---|---|---|---|---|
| Quantizer | 2.0 | 2.25 | 2.5 | 2.75 | 3.0 | 3.25 | 3.5 | 3.75 | 4.0 | 4.25 |
| NF w/ FLUTE [11, 20] | – | – | – | – | – | 2.33× | – | – | – | 2.20× |
| QTIP [50] | 2.91× | – | – | – | 2.49× | – | – | – | 2.23× | – |
| Ours-NUQ-TC | 3.64× | – | – | – | 2.97× | – | – | – | 2.57× | – |
| Ours-NUQ-CC | **3.70×** | – | – | – | 3.07× | – | – | – | **2.61×** | – |
| Ours-VQ-TC | 3.63× | – | 3.24× | – | 2.97× | – | 2.69× | – | 2.57× | – |
| Ours-VQ-CC | 3.69× | – | **3.36×** | – | **3.07×** | – | **2.82×** | – | 2.60× | – |
| Ours-TCQ-TC | 3.57× | **3.23×** | 3.13× | **2.95×** | 2.96× | **2.72×** | 2.70× | **2.61×** | 2.59× | **2.37×** |

| Decoding latency speedup compared to FP16 (batch size = 8) | | | | | | | | | |
|---|---|---|---|---|---|---|---|---|---|
| Quantizer | 2.0 | 2.25 | 2.5 | 2.75 | 3.0 | 3.25 | 3.5 | 3.75 | 4.0 | 4.25 |
| NF w/ FLUTE [11, 20] | – | – | – | – | – | 2.28× | – | – | – | 2.13× |
| QTIP [50] | 0.74× | – | – | – | 0.55× | – | – | – | 0.47× | – |
| Ours-NUQ-TC | **3.28×** | – | – | – | **2.78×** | – | – | – | 2.46× | – |
| Ours-NUQ-CC | 2.80× | – | – | – | 2.56× | – | – | – | 2.11× | – |
| Ours-VQ-TC | **3.28×** | – | **2.87×** | – | 2.74× | – | 2.34× | – | **2.47×** | – |
| Ours-VQ-CC | 2.94× | – | 2.72× | – | 2.66× | – | 2.46× | – | 2.37× | – |
| Ours-TCQ-TC | 3.16× | **2.92×** | 2.80× | **2.75×** | 2.74× | **2.56×** | **2.47×** | **2.49×** | 2.46× | **2.27×** |

**Trellis-coded quantization.** TCQ is known as a sophisticated Gaussian source quantizer, achieving near-optimal distortion performance [19]. Recently, QTIP introduced optimized CUDA kernels for the bitshift variant of TCQ [36, 50], which encodes each high-dimensional real-valued vector $\mathbf{v}$ into a binary bitshift representation $\mathbf{r}$:

$$\hat{\mathbf{v}}[i \cdot V : (i+1) \cdot V] = \text{LUT}\left(\mathbf{r}[i \cdot s : i \cdot s + L]\right),$$

where each $V$-dimensional subvector is represented by a sliding bit-window of length $L$, shifted by $s$ bits at each step. This encoding achieves an effective fractional bitwidth of $s/V$. While previous TCQ kernel (QTIP) supported integer bitwidths (*e.g.*, 2, 3, 4 bits with shifts $s = 4, 6, 8$ for $V = 2$) and were limited to single batch, we significantly expand practical applicability by introducing fractional bitwidth support (*e.g.*, 1.5, 2.5, 3.5, 4.5, 5.0 bits corresponding to shifts $s = 3, 5, 7, 9, 10$ for $V = 2$) and optimized kernels supporting inference at batch sizes up to 8. For constructing the LUT, we follow the protocol of QTIP, which is also based on $k$-means clustering of Gaussian samples.

Furthermore, we introduce *half-TCQ*, a simple extension enabling quantization at intermediate fractional bitwidth intervals. Specifically, given a weight $W \in \mathbb{R}^{d_{\text{in}} \times d_{\text{out}}}$, half-TCQ partitions the matrix row-wise and applies different bitwidth quantization to each partition. For example, to achieve 2.75 bit quantization, half-TCQ quantizes $W[: d_{\text{in}}/2]$ at 2.5 bits and the remaining half $W[d_{\text{in}}/2 :]$ at 3 bits. To preserve computational efficiency, we implement a dedicated CUDA kernel that performs fused dequantization and matrix multiplication for half-TCQ in a single kernel call. As illustrated in Figure 2, TCQ-based schemes, including half-TCQ, consistently achieve quantization error close to theoretical lower bounds, outperforming simpler quantizers. For a more detailed explanation of these quantizers, including quantization algorithms, please refer to Appendix C.

### 3.3 Implementation details for efficiency

**Reducing rotation overhead.** Incoherence processing rotates weights along both input and output dimensions ($W \rightarrow RWR'$) with per-tensor scaling, yielding approximately standard Gaussian distributions. However, each rotation also requires rotating activations online during inference (*e.g.*, $X \rightarrow XR$), incurring significant computational overhead [6, 49, 50]. We reduce this overhead by rotating weights only along the input dimension ($W \rightarrow RW$) and applying per-output-channel scaling, normalizing each rotated column to an approximately standard Gaussian distribution. Additionally, we share rotation matrices among linear layers with identical inputs (*e.g.*, query/key/value or gate/up projections in Transformer blocks). Combining these techniques reduces the number of online rotations per Transformer block from 14 to 4.

**Kernel implementation.** We implement two types of CUDA kernels optimized for different inference scenarios: (i) Tensor Core-based kernels and (ii) CUDA Core-based kernels. Our Tensor Core-based kernels, supporting TCQ, NUQ, and VQ, extend the single batch implementation from QTIP,

which leverages warp-level `mma` (matrix-multiply-accumulate) instructions [38]. Integrating efficient dequantization logic into this framework involves non-trivial engineering efforts, including the precise mapping of quantized weights to `mma` instruction fragments. Overall, our implementation extends kernel support from integer bitwidths (2, 3, 4 bits) to fractional bitwidths (1.5, 2.5, 3.5, 4.5, 5.0 bits). To further minimize overhead at larger batch sizes, we traverse each quantized weight exactly once, directly loading and performing register-level dequantization without intermediate storage. Input activations are cached in shared memory for efficient reuse across multiple weight multiplications, significantly improving inference efficiency.

Our CUDA Core-based kernels, supporting NUQ and VQ, extend the Any-Precision LLM kernels [40], originally designed for NUQ. Specifically, we replace bit-plane encoding with simpler bit-packing encoding to simplify the dequantization process. Additionally, we incorporate Any-precision's table lookup merging technique into both Tensor Core and CUDA Core-based NUQ kernels for bitwidths 2, 3, and 4, further reducing dequantization overhead. Table 1 summarizes the supported kernel implementations and bitwidths for quantization schemes in Q-Palette. For transparency and reproducibility, the full implementations of both kernel types are included in our code release, and additional kernel analysis is provided in Appendix D.

Table 2 summarizes the end-to-end decoding-latency speedup for LLaMA 3.1 8B models quantized at various bitwidths (2.0-4.25 bits) using quantizers in Q-Palette, compared against two baselines: NormalFloat with FLUTE kernels and QTIP [11, 20, 50]. Our quantizers consistently deliver superior inference speed while supporting a wide range of finer-grained fractional bit quantization. For smaller batch sizes (batch size = 1), CUDA Core-based kernels typically outperform Tensor Core-based kernels, whereas Tensor Core-based kernels often provide better latency at larger batch sizes (batch size = 8). Although TCQ incurs slightly higher latency overhead compared to NUQ and VQ, our TCQ quantizers still achieve significantly faster decoding speed compared to baseline quantizers, clearly demonstrating practical efficiency improvements for real-world inference workloads. Note that our TCQ quantizers extend QTIP and, at batch size 1, primarily differs by supporting a wider range of bitwidths and reducing the number of online Hadamard transforms, thereby lowering rotation cost and improving decoding-latency speedup, *e.g.*, from $2.91\times$ to $3.57\times$ at 2-bit.

## 4 Mixed-scheme quantization with Q-Palette

### 4.1 Mixed-scheme quantization under resource constraints

The memory-constrained MSQ formulation introduced in Section 2 naturally generalizes to broader resource constraints such as inference latency. Following prior one-shot mixed-precision quantization frameworks [14, 7, 35], we formulate resource-constrained MSQ as MCKP:

$$\underset{P_{lq}\in\{0,1\}}{\text{minimize}} \sum_{l=1}^{L}\sum_{q=1}^{|\mathcal{Q}|} P_{lq}\cdot\ell_{lq} \qquad (3)$$

$$\text{subject to} \sum_{q=1}^{|\mathcal{Q}|} P_{lq} = 1, \quad \forall 1\leq l\leq L,$$

$$\sum_{l=1}^{L}\sum_{q=1}^{|\mathcal{Q}|} P_{lq}\cdot c_{lq} \leq C,$$

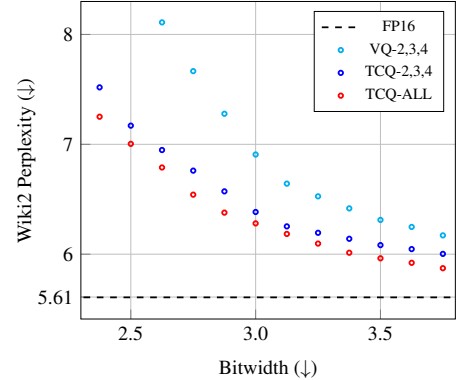

Figure 3: Performance comparison of memory-constrained MSQ for different quantizer sets in Q-Palette on LLaMA 3.1-8B.

where the loss term $\ell_{lq}$ denotes the estimated loss in performance incurred by selecting quantizer $Q_q$ for layer $l$, the cost term $c_{lq} \triangleq \text{cost}(Q_q; W_l)$ represents the profiled resource cost (*e.g.*, memory or latency), and the total resource constraint is denoted by $C$. The loss term $\ell_{lq}$ can be instantiated in various ways depending on available information. For example, in data-free settings, we approximate the loss term as $\ell_{lq} = a_l \cdot \text{err}(Q_q; W_l)$, as in Equation (1), where $a_l$ is a layer-wise sensitivity coefficient computed following the protocol of HIGGS [35]. We estimate $\text{err}(Q_q; W_l)$ using the precomputed distortion of $Q_q$, obtained by quantizing a random Gaussian matrix. This enables fast estimation of loss terms in data-free scenarios without fully realizing the quantization pipeline.

While a dynamic programming algorithm exists for solving MCKP [43], we simply use the SCIP solver provided by Google OR-Tools for flexibility and ease of implementation [42]. Please refer to Appendix E for additional details on loss term computation and cost profiling.

We illustrate the effectiveness of Q-Palette in Figure 3 by comparing memory-constrained MSQ results across different quantizer subsets within Q-Palette. Specifically, *TCQ-ALL* includes all TCQ quantizers available in Q-Palette, while *TCQ-2,3,4* and *VQ-2,3,4* reflect integer-bitwidth quantizers supported in QTIP and HIGGS, respectively [50, 35]. Although QTIP is originally a single-scheme baseline, our constructed integer-bitwidth subset *TCQ-2,3,4* serves as a reasonable reference to evaluate the benefit of broadened quantizer support. The results clearly highlight the advantage of *TCQ-ALL*, demonstrating that the broadened quantizer support provided by Q-Palette consistently yields superior performance.

## 4.2   Fusion-aware mixed-scheme quantization

*Layer fusion* is a widely used optimization for accelerating inference speed of DNN models [26, 12]. Within each Transformer block, certain linear layers, such as {query, key, value} projections or {up, gate} projections, share the same input and can be fused into a single linear layer. For example, instead of separately computing $XW_q$, $XW_k$, and $XW_v$, we can concatenate the weight matrices and compute $X(W_q \oplus W_k \oplus W_v)$, followed by splitting the output (see the right side of Figure 1). Layer fusion can reduce the number of kernel launches and memory accesses, thereby providing further opportunities for inference speedup.

We propose *fusion-aware MSQ*, a novel MSQ framework that jointly optimizes quantization with the additional design dimension of layer fusion. Fusion-aware MSQ simultaneously determines 1) how to group layers for fusion and 2) which quantizer to assign to each fused group. Whereas standard MSQ (Equation (3)) introduces one binary decision variable per (layer, quantizer) pair, fusion-aware MSQ instead defines one binary variable per (fusible layer group, quantizer) pair. Here, a fusible layer group is a set of layers sharing the same input. For generic Transformer models, we can write the set of all fusible layer groups for each Transformer block $b$ as

$$\mathcal{G}_b = \left\{ \{q_b\}, \{k_b\}, \{v_b\}, \{q_b, k_b\}, \{q_b, v_b\}, \{k_b, v_b\}, \{q_b, k_b, v_b\}, \{o_b\}, \{u_b\}, \{g_b\}, \{u_b, g_b\}, \{d_b\} \right\}.$$

The overall set of fusible layer groups is $\mathcal{G} = \bigcup_{b=1}^{B} \mathcal{G}_b$ where $B$ is the number of Transformer blocks. For each $g \in \mathcal{G}$ and quantizer $Q_q \in \mathcal{Q}$, we introduce a binary variable $P_{gq} \in \{0, 1\}$ indicating that all layers in $g$ are fused and quantized by $Q_q$.

The fusion-aware MSQ problem is formulated as:

$$\underset{P_{gq} \in \{0,1\}}{\text{minimize}} \quad \sum_{g \in \mathcal{G}} \sum_{q=1}^{|\mathcal{Q}|} P_{gq} \cdot \sum_{l \in g} \ell_{lq} \tag{4}$$

$$\text{subject to} \quad \sum_{g \in \mathcal{G}:l \in g} \sum_{q=1}^{|\mathcal{Q}|} P_{gq} = 1, \quad \forall l \in \bigcup_{b=1}^{B} \{q_b, k_b, v_b, o_b, u_b, g_b, d_b\}, \tag{C1}$$

$$\sum_{g \in \mathcal{G}} \sum_{q=1}^{|\mathcal{Q}|} P_{gq} \cdot c_{gq} \leq C, \tag{C2}$$

where $\ell_{lq}$ is the loss term, $c_{gq}$ represents the profiled cost (*e.g.*, latency) of the fused layer corresponding to the group $g$ quantized by $Q_q$.

To ensure valid solutions, two constraints are imposed. *Exclusive assignment* (C1): every layer must belong to exactly one active (group, quantizer) pair; among all fusible groups $g$ that contain a given layer $l$, only one associated variable $P_{gq}$ can be 1. *Resource constraint* (C2): the total profiled cost of all activated groups must not exceed the resource budget $C$.

This formulation explicitly captures latency improvements enabled by fusion, thus providing improved accuracy-latency trade-offs compared to the MSQ formulation that neglects layer fusion (see Figure 1). Since both the objective and constraints are linear in $P_{gq}$, Equation (4) is also an ILP. We solve this ILP using the SCIP solver in OR-Tools [42]. Note that, compared to non-fusion-aware MSQ (Equation (3)), fusion-aware MSQ introduces $1.71\times$ more decision variables while maintaining the same number of constraints.

Table 3: Data-free quantization results on LLaMA 3 models for various bitwidths.

| Method | LLaMA 3.1-8B | | | LLaMA 3.2-1B | | | LLaMA 3.2-3B | | |
|---|---|---|---|---|---|---|---|---|---|
| | Bits ($\downarrow$) | Wiki2 ($\downarrow$) | Acc ($\uparrow$) | Bits ($\downarrow$) | Wiki2 ($\downarrow$) | Acc ($\uparrow$) | Bits ($\downarrow$) | Wiki2 ($\downarrow$) | Acc ($\uparrow$) |
| FP16 | 16.00 | 5.61 | 69.3 | 16.00 | 8.64 | 55.9 | 16.00 | 6.98 | 63.7 |
| Data-free QTIP | 3.00 | 6.81 | **66.9** | 3.00 | 13.35 | 49.0 | 3.00 | 8.89 | 58.2 |
| Ours-TCQ-3 | 3.00 | **6.78** | 66.0 | 3.00 | **12.59** | **50.7** | 3.00 | **8.67** | **60.3** |
| Ours-MSQ-Mem | 3.00 | **6.28** | **67.5** | 3.00 | **10.51** | **53.2** | 3.00 | **7.81** | **61.7** |
| NF | 3.25 | 7.70 | 64.3 | 3.25 | 17.73 | 46.8 | 3.25 | 10.06 | 59.3 |
| HQQ | 3.25 | 8.29 | 63.2 | 3.25 | 26.42 | 42.9 | 3.25 | 11.68 | 54.2 |
| HIGGS | 3.25 | 6.64 | 66.4 | 3.25 | 12.19 | 51.1 | 3.25 | 8.67 | 60.1 |
| Ours-TCQ-3.25 | 3.25 | **6.48** | **66.4** | 3.25 | **11.30** | **51.8** | 3.25 | **8.15** | **61.0** |
| HIGGS-MSQ | 3.25 | 6.39 | 66.7 | 3.25 | 11.08 | 52.5 | 3.25 | 8.01 | 61.1 |
| Ours-MSQ-Mem | 3.25 | **6.10** | **67.6** | 3.25 | **10.00** | **53.7** | 3.25 | **7.60** | **61.9** |
| NF | 4.02 | 6.22 | 67.8 | 4.02 | 10.70 | 52.9 | 4.02 | 7.82 | 62.1 |
| HQQ | 4.02 | 6.52 | 67.5 | 4.02 | 13.47 | 51.4 | 4.02 | 8.67 | 60.2 |
| HIGGS | 4.02 | 5.98 | **68.7** | 4.02 | 9.64 | 53.6 | 4.02 | 7.46 | 62.3 |
| Data-free QTIP | 4.00 | 5.94 | 68.4 | 4.00 | 9.53 | **54.9** | 4.00 | 7.41 | 62.8 |
| Ours-TCQ-4 | 4.00 | **5.92** | 68.2 | 4.00 | **9.45** | 54.3 | 4.00 | **7.37** | **63.4** |
| HIGGS-MSQ | 4.00 | 5.91 | 68.3 | 4.00 | 9.52 | 55.0 | 4.00 | 7.40 | 62.2 |
| Ours-MSQ-Mem | 4.00 | **5.81** | **69.0** | 4.00 | **9.14** | **55.2** | 4.00 | **7.22** | **63.2** |

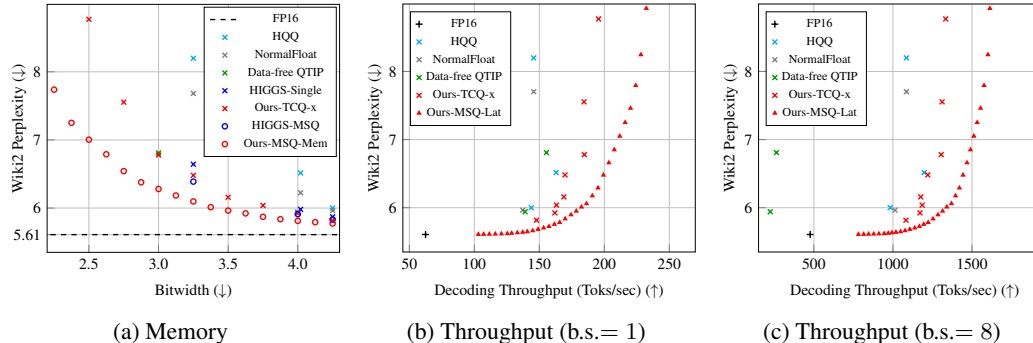

| (a) Memory | (b) Throughput (b.s.$= 1$) | (c) Throughput (b.s.$= 8$) |
|---|---|---|

Figure 4: Performance trade-offs of quantized LLaMA 3.1-8B models under different constraints in the data-free setting on an RTX 4090 GPU: (a) memory constraint; (b) latency constraint (single batch); (c) throughput evaluation (batch size $= 8$) of the quantized models in (b).

# 5 Experiments

We evaluate the quantization performance of our methods against baselines on the LLaMA 3 series (LLaMA 3.1-8B, 70B, 3.2-1B, 3B), LLaMA 2 series (LLaMA 2-7B, 13B), and Qwen 2.5-7B [18, 48, 55]. For the data-free scenario, we consider single-scheme quantization baselines—HQQ (uniform), NormalFloat (NUQ), HIGGS-Single (VQ), and data-free QTIP (TCQ)—as well as the MSQ baseline HIGGS-MSQ [2, 11, 50, 35]. For the data-aware scenario, we use QTIP as the baseline. For all experiments, both our methods and baselines are evaluated strictly in the PTQ setting, without any retraining. Performance is measured primarily via WikiText2 perplexity and average zero-shot accuracy across ARC-easy, ARC-challenge, HellaSwag, PiQA, and WinoGrande [37, 8, 56, 3, 44]. For latency evaluations, we use Gemlite kernels for HQQ at higher bitwidths (4.02, 4.25 bits), and FLUTE kernels for NormalFloat (3.25, 4.25 bits) and HQQ at 3.25 bits [22, 20]. Because QTIP supports only single batch inference, we simulate larger batch sizes by repeated kernel invocation.

We denote *Ours-TCQ-x* as single-scheme quantization using TCQ-$x$ from Q-Palette, *Ours-MSQ-Mem* as memory-constrained MSQ (Section 4.1) using Q-Palette's TCQ quantizers, and *Ours-MSQ-Lat* as latency-constrained fusion-aware MSQ (Section 4.2) using all Q-Palette quantizers with Tensor-Core kernels. Please refer to Appendix G for additional results on other models, ablation studies, and detailed experimental settings.

Table 4: Data-aware quantization results on LLaMA 2 models (throughput on an RTX4090 GPU).

| | | LLaMA 2 7B | | | | | LLaMA 2 13B | | | |
| | | | | Throughput (Toks/s) | | | | | Throughput (Toks/s) | |
| Method | Bits | Wiki2 ($\downarrow$) | Acc ($\uparrow$) | $B=1$ | $B=8$ | Bits | Wiki2 ($\downarrow$) | Acc ($\uparrow$) | $B=1$ | $B=8$ |
|---|---|---|---|---|---|---|---|---|---|---|
| FP16 | 16.00 | 5.12 | 64.9 | 71 | 527 | 16.00 | 4.57 | 67.9 | OOM | OOM |
| QTIP | 2.00 | 6.84 | 58.9 | 209 | 386 | 2.00 | 5.62 | 63.6 | 131 | 154 |
| Ours-MSQ-Mem | 2.00 | **6.47** | **60.3** | **272** | **1684** | 2.00 | **5.35** | **64.2** | **152** | **928** |
| QTIP | 3.00 | 5.39 | 63.3 | 184 | 304 | 3.00 | 4.76 | **67.0** | 110 | 153 |
| Ours-MSQ-Mem | 3.00 | **5.34** | **63.9** | **224** | **1489** | 3.00 | **4.74** | 67.0 | **126** | **738** |

## 5.1 Data-free quantization results

Table 3 summarizes data-free quantization performance on LLaMA 3 models. *Ours-TCQ-x* consistently outperforms all single-scheme baselines in WikiText2 perplexity and achieves competitive zero-shot accuracy. Notably, *Ours-MSQ-Mem* surpasses all baseline methods, clearly demonstrating the effectiveness of Q-Palette. Figure 4a evaluates *Ours-MSQ-Mem* across a broader memory range (2.25-4.25 bits). Our method achieves Pareto-dominant performance, significantly outperforming baseline methods. Remarkably, our 2.875-bit model achieves comparable WikiText2 perplexity to the 3.25-bit HIGGS-MSQ model, resulting in a $1.13\times$ higher compression ratio and superior perplexity. Figures 4b and 4c compare the throughput-perplexity trade-offs of *Ours-MSQ-Lat* and *Ours-TCQ-x* against baseline methods. Both variants achieve significant throughput improvements over baseline methods, substantially expanding the Pareto frontier in both batch sizes 1 and 8.

## 5.2 Data-aware quantization results

We further evaluate our methods in the data-aware setting by comparing our *Ours-MSQ-Mem* approach against the state-of-the-art QTIP baseline (without retraining) on the LLaMA 2-7B and 13B models. For this setting, we utilize the same proxy Hessian used in QTIP during the quantization and compute the loss term $\ell_{lq}$ for our MSQ as the actual validation perplexity degradation induced by quantizing the weight $W_l$ using the quantizer $Q_q$. As summarized in Table 4, our method consistently achieves superior perplexity and zero-shot accuracy compared to the baseline. Additionally, our optimized kernels achieve over $4\times$ throughput improvements at batch size 8 for both LLaMA 2 models at both 2 and 3 bits, demonstrating the practical benefits of our optimized kernel for batch size 8.

# 6 Related works

**Incoherence processing.** Previous methods for handling outliers in LLM quantization have primarily relied on heuristic techniques [28, 54, 29]. Recently, a theoretically grounded approach, incoherence processing, has been introduced to systematically address weight irregularities [6]. Incoherence processing applies rotation matrices to weight matrices prior to quantization, significantly suppressing outliers and transforming distributions into approximately Gaussian forms [6, 1, 45]. This Gaussianization enables the use of sophisticated Gaussian quantizers such as lattice vector quantization [49] and trellis-coded quantization [50]. However, current implementations support efficient kernels only for limited integer bitwidths and small batch sizes, constraining their practicality, a limitation that our proposed Q-Palette directly addresses by introducing fractional-bit quantizers and optimized CUDA kernels with broader batch size support.

More recent rotation-based approaches further enhance quantization performance by applying learned matrix transforms such as scaling or affine transformations [32, 30, 23, 47]. However, these methods mainly target weight-activation quantization and require calibration data to learn the transforms, whereas our work focuses on weight-only PTQ, which remains applicable even in data-free settings and is particularly suited for memory-bound, small-batch inference.

**Other weight-only post-training quantization methods.** Several simpler PTQ methods prioritize computational and implementation efficiency. HQQ employs data-free uniform quantization via half-quadratic optimization [2]. NormalFloat constructs lookup tables for non-uniform scalar quantization using Gaussian quantiles [11]. FLUTE offers state-of-the-art CUDA kernels for LUT-based non-

uniform quantizers with per-group scaling [20]. Despite their efficiency, these approaches generally incur higher quantization errors compared to sophisticated quantizers such as TCQ.

**Mixed-precision and mixed-scheme quantization.** Mixed-precision quantization (MPQ) optimizes layer-wise bit allocation under given constraints [53]. For vision models, HAQ and HAWQ-V2 introduced surrogate objectives based on second-order information for MPQ [13, 14]. Chen et al. generalized these approaches by explicitly incorporating diverse resource constraints, such as latency, and formulated the problem as an MCKP [7]. Recently, HIGGS introduced the linearity theorem, a data-free linear surrogate specifically tailored for LLM quantization [35]. Building upon these works and drawing insights from compiler optimization research [26, 12], we propose a novel fusion-aware mixed-scheme quantization framework that jointly optimizes quantizer selection and layer fusion decisions, achieving superior accuracy-latency trade-offs.

# 7  Conclusion

In this paper, we have investigated weight-only PTQ as a solution for compressing LLMs, particularly beneficial for memory-bound inference tasks with small batch sizes. Considering that irregular weight distributions in LLMs have complicated quantization, we leveraged recent rotation-based methods that Gaussianize weight distributions, enabling a theoretical analysis of optimal bit allocation. Based on this perspective, we derived an information-theoretically optimal bit allocation strategy under fixed bit budgets, demonstrating that fine-grained fractional-bit quantizers closely approaching the Gaussian distortion-rate bound are essential for achieving near-optimal quantization efficiency. To translate this theoretical finding into practical benefits, we introduced Q-Palette, a versatile suite of fractional-bit quantizers, from sophisticated trellis-coded quantization schemes offering near-optimal distortion to simpler vector and scalar quantizers optimized for fast inference, each efficiently implemented with optimized CUDA kernels across a wide range of bitwidths. We further integrated Q-Palette into an MSQ framework, proposing a novel fusion-aware MSQ approach that jointly optimizes quantizer selection and layer fusion decisions under given resource constraints, effectively improving inference latency. Experimental evaluations validated that our MSQ framework with Q-Palette and fusion-aware optimization consistently outperforms existing baseline methods, achieving superior accuracy-memory and accuracy-latency trade-offs on LLaMA 2 and LLaMA 3 models.

## Impact statement

Q-Palette introduces a versatile set of quantizers with broad fractional-bitwidth support, which can serve as a foundational building block for evaluating and developing MSQ algorithms. Q-Palette's quantizers are usable in data-free scenarios, offering off-the-shelf applicability like NormalFloat and HQQ [11, 20, 2], which lowers the barrier for practitioners lacking calibration data. Importantly, Q-Palette supports a wide spectrum of performance-efficiency trade-offs, enabling practitioners to select quantization configurations that best match their specific deployment workloads. This adaptability is valuable for real-world applications where resource constraints and performance requirements vary significantly. Moreover, our results challenge the misconception that sophisticated quantizers such as TCQ are computationally prohibitive for practical use beyond batch size 1 [35]. We demonstrate that TCQ achieves efficient decoding speeds for batch sizes up to 8, making it practically suitable for edge-device workloads. By correcting this misunderstanding, our work may encourage further investigation into TCQ and other quantizers previously considered computationally expensive.

## Acknowledgements and Disclosure of Funding

We would like to thank Jinuk Kim for insightful discussions and helpful feedback on this work. This work was supported by Samsung Electronics Co., Ltd. (IO250418-12669-01), Mobile eXperience (MX) Business, Samsung Electronics Co., Ltd., Institute of Information & Communications Technology Planning & Evaluation (IITP) grant funded by the Korea government (MSIT) [No. RS-2020-II200882, (SW STAR LAB) Development of deployable learning intelligence via self-sustainable and trustworthy machine learning, No. RS-2021-II211343, Artificial Intelligence Graduate School Program (Seoul National University)], and the National Research Foundation of Korea (NRF) grant funded by the Korea government (MSIT) (No. RS-2024-00354036). Hyun Oh Song is the corresponding author.

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

# A Optimal bitwidth proof

In this section, we formally derive the optimal bit allocation result stated in the main paper. Under the assumption that weight matrices have been Gaussianized through incoherence processing, the quantization problem can be viewed as a Gaussian source coding problem. We recall the memory-constrained mixed-scheme quantization (MSQ) formulation as:

$$
\begin{aligned}
\underset{P_{lq} \in \{0,1\}}{\text{minimize}} \quad & \sum_{l=1}^{L} a_l \left( \sum_{q=1}^{|\mathcal{Q}|} P_{lq} \cdot \text{err}(Q_q; W_l) \right) \\
\text{subject to} \quad & \sum_{q=1}^{|\mathcal{Q}|} P_{lq} = 1, \quad \forall 1 \le l \le L, \\
& \sum_{l=1}^{L} \sum_{q=1}^{|\mathcal{Q}|} P_{lq} \cdot \text{bit}(Q_q; W_l) d_l^{\text{in}} d_l^{\text{out}} \le M,
\end{aligned}
\tag{1}
$$

where $a_l$ is the empirically estimated sensitivity coefficient for layer $l$, $\text{err}(Q; W_l)$ is the normalized quantization error of layer $l$, $Q_q$ denotes a candidate quantizer, $\text{bit}(Q_q; W_l)$ is the average number of bits per weight component for the weight matrix $W_l$ quantized by $Q_q$, $P_{lq} \in \{0, 1\}$ is a binary indicator selecting quantizer $Q_q$ for layer $l$, and $M$ denotes the total memory budget (in bits) allocated for quantized model [46, 35, 7].

We recall that classical rate-distortion theory provides a fundamental lower bound on the expected quantization error for Gaussian sources: $\mathbb{E}[\text{err}(Q)] \ge 2^{-2\text{bit}(Q)}$ [10]. Further assuming we have access to ideal Gaussian quantizers capable of exactly achieving this theoretical distortion bound $\mathbb{E}[\text{err}(Q)] = 2^{-2\text{bit}(Q)}$ at any fractional bitwidth $b_l \ge \eta$, the memory-constrained MSQ (problem (1)) can again be written as the continuous optimization problem:

$$
\begin{aligned}
\underset{b_l \ge \eta}{\text{minimize}} \quad & \sum_{l=1}^{L} a_l 2^{-2b_l} \\
\text{subject to} \quad & \sum_{l=1}^{L} b_l d_l^{\text{in}} d_l^{\text{out}} \le M,
\end{aligned}
\tag{2}
$$

where $b_l$ is the fractional bitwidth allocated to layer $l$, and $\eta > 0$ is a minimum bitwidth threshold introduced to avoid degenerate cases such as assigning 0-bit to a layer. Here, we replace the actual quantization error $\text{err}(Q)$ with its expectation $\mathbb{E}[\text{err}(Q)]$. We empirically justify this replacement by demonstrating extremely low variance in quantization errors for typical weight matrix dimensions encountered in LLMs (see Table 5). Additionally, we assume that the sensitivity coefficients $a_l$ are non-negative ($a_l \ge 0$), a reasonable assumption given that pretrained weights typically represent local optima. Given this simplified optimization problem, we now derive the closed-form solution for the optimal fractional bit allocation.

**Theorem 3.1** (Optimal bit allocation with ideal Gaussian quantizers). *If the budget $M$ is feasible, i.e., $M \ge \eta \sum_{l=1}^{L} d_l^{in} d_l^{out}$, then the optimal fractional bit allocation $\{b_l^*\}$ for problem (2) is given by*

$$
b_l^* = \max \left\{ \eta, \frac{1}{2\ln(2)} \left( \ln \frac{a_l}{d_l^{\text{in}} d_l^{\text{out}}} \right) + C \right\}, \quad \forall 1 \le l \le L,
$$

*for the constant $C$ that satisfies the memory constraint $\sum_l b_l^* d_l^{\text{in}} d_l^{\text{out}} = M$.*

*Proof.* Let's start by formulating the Lagrangian function for problem (2), explicitly including the constraint $b_l \ge \eta$ via Lagrange multipliers $\mu_l \ge 0$ and the budget constraint via $\lambda \ge 0$:

$$
\mathcal{L}(\{b_l\}, \lambda, \{\mu_l\}) := \sum_{l=1}^{L} a_l 2^{-2b_l} + \lambda \left( \sum_{l=1}^{L} b_l d_l^{\text{in}} d_l^{\text{out}} - M \right) - \sum_{l=1}^{L} \mu_l (b_l - \eta).
$$

Table 5: Empirical distortion statistics for quantizing random Gaussian matrices ($4096 \times 4096$) with Q-Palette quantizers over 32 trials.

| Quantizer | Mean distortion | Std. deviation |
|---|---|---|
| Ours-TCQ-2 | 0.07101 | 8.36E-06 |
| Ours-NUQ-2 | 0.11747 | 4.24E-05 |
| Ours-VQ-2 | 0.10857 | 2.93E-05 |

By differentiating the Lagrangian with respect to $b_l$ and setting it equal to zero to find the stationary points, we have:

$$\frac{\partial \mathcal{L}}{\partial b_l} = -2\ln(2)\,a_l 2^{-2b_l} + \lambda d_l^{\text{in}} d_l^{\text{out}} - \mu_l = 0.$$

Since, for all $l$, $\mu_l \geq 0$ and complementary slackness requires $\mu_l(b_l^* - \eta) = 0$ [4], we have two cases:
**Case 1:** If $b_l^* > \eta$, complementary slackness implies $\mu_l = 0$, and thus:

$$2^{-2b_l^*} = \frac{\lambda d_l^{\text{in}} d_l^{\text{out}}}{2\ln(2)\,a_l}.$$

Taking logarithms on both sides and rearranging terms explicitly, we obtain:

$$b_l^* = \frac{1}{2\ln(2)}\left(\ln\frac{a_l}{d_l^{\text{in}} d_l^{\text{out}}}\right) + \frac{1}{2\ln(2)}\left(\ln(2\ln(2)) - \ln(\lambda)\right) = \frac{1}{2\ln(2)}\left(\ln\frac{a_l}{d_l^{\text{in}} d_l^{\text{out}}}\right) + C(\lambda).$$

where, the constant term $C(\lambda)$ is defined as $C(\lambda) := \frac{1}{2\ln(2)}(\ln(2\ln(2)) - \ln(\lambda))$.

**Case 2:** If $b_l^* = \eta$, we directly have:

$$b_l^* = \eta.$$

Combining these cases yields the optimal fractional bit allocation:

$$b_l^* = \max\left\{\eta, \frac{1}{2\ln(2)}\left(\ln\frac{a_l}{d_l^{\text{in}} d_l^{\text{out}}}\right) + C(\lambda)\right\},$$

where the constant $C(\lambda)$ is chosen such that the memory constraint

$$\sum_{l=1}^{L} b_l^* d_l^{\text{in}} d_l^{\text{out}} \leq M,$$

is tight (*i.e.*, equality holds). This equality condition emerges naturally, as the objective function (2) is non-increasing in $b_l$ due to the non-negativity assumption ($a_l \geq 0$). Therefore, increasing $C(\lambda)$ until the constraint is exactly met cannot worsen the objective, completing the proof. □

To empirically validate our approximation of quantization errors by their expectation, we quantized random Gaussian matrices multiple times and observed consistently low variance in the quantizataion errors (distortion). Specifically, we performed quantization on 32 random standard Gaussian matrices of shape $(4096, 4096)$, consistent with the LLaMA 3.1-8B self-attention query projection layer. Table 5 reports the mean and standard deviation of the normalized quantization error ($\|\hat{W} - W\|_2^2 / \|W\|_2^2$) values. The results demonstrate low variance, supporting our assumption.

## B  Analysis of the quantization optimality gap

In practice, we typically have access only to a finite set of quantizers $\mathcal{Q} = \{Q_1, \ldots, Q_N\}$, which may not achieve the theoretical distortion-rate optimality. Under this constraint, the original memory-constrained MSQ problem (1) can still be solved, but the resulting solution may deviate from the optimal solution derived under the assumption of ideal Gaussian quantizers (Theorem 3.1). In this section, we formally analyze this quantization optimality gap.

Table 6: Optimality gap analysis for different quantizer sets on LLaMA 3.1-8B. TCQ-ALL includes all fractional TCQ bitwidths from 1.5 to 5.0 in Q-Palette.

| Quantizer pool | Bitwidth | Distortion gap ($\downarrow$) | Bit allocation gap ($\downarrow$) | Total gap ($\downarrow$) | Surrogate objective ($\downarrow$) |
|---|---|---|---|---|---|
| VQ-2,3,4 | 3.25 | 0.0586 | 0.0130 | 0.0716 | 0.1219 |
| TCQ-2,3,4 | 3.25 | 0.0198 | 0.0129 | 0.0327 | 0.0830 |
| TCQ-ALL | 3.25 | 0.0145 | 0.0023 | 0.0168 | 0.0671 |
| Ideal Gaussian quantizer | 3.25 | 0 | 0 | 0 | 0.0503 |
| VQ-2,3,4 | 2.50 | 0.1282 | 0.0178 | 0.1460 | 0.2883 |
| TCQ-2,3,4 | 2.50 | 0.0306 | 0.0178 | 0.0484 | 0.1907 |
| TCQ-ALL | 2.50 | 0.0260 | 0.0034 | 0.0294 | 0.1717 |
| Ideal Gaussian quantizer | 2.50 | 0 | 0 | 0 | 0.1423 |

Let $Q_l^*$ denote the optimal quantizer selected from the quantizer set $\mathcal{Q}$ for each $l$, obtained by solving problem (1). Then, the quantization optimality gap, defined as the difference in the objective values between the practical optimal solution and the theoretically optimal fractional bitwidth solution $b_l^*$ (derived in Theorem 3.1), can be expressed as:

$$\sum_{l=1}^{L} a_l \left( \mathrm{err}(Q_l^*; W_l) - 2^{-2b_l^*} \right). \tag{5}$$

We decompose the total gap into two intuitive terms, the distortion gap and the bit allocation gap as follows:

$$\underbrace{\sum_{l=1}^{L} a_l \left( \mathrm{err}(Q_l^*) - 2^{-2b_l^*} \right)}_{\text{Total gap}} = \underbrace{\sum_{l=1}^{L} a_l \left( \mathrm{err}(Q_l^*) - 2^{-2\mathrm{bit}(Q_l^*)} \right)}_{\text{Distortion gap}} + \underbrace{\sum_{l=1}^{L} a_l \left( 2^{-2\mathrm{bit}(Q_l^*)} - 2^{-2b_l^*} \right)}_{\text{Bit allocation gap}},$$

where we abbreviate $\mathrm{err}(Q_l^*; W_l)$ by $\mathrm{err}(Q_l^*)$ and $\mathrm{bit}(Q_l^*; W_l)$ by $\mathrm{bit}(Q_l^*)$ for the simplicity.

Due to classical rate-distortion theory and the optimality of $b_l^*$ as the solution of the continuous optimization problem (2), each term in this decomposition is non-negative. Specifically, the first term, $\left( \mathrm{err}(Q_l^*) - 2^{-2\mathrm{bit}(Q_l^*)} \right)$, quantifies how closely each practical quantizer $Q_l^*$ approaches the theoretical Gaussian distortion bound. The second term, $\left( 2^{-2\mathrm{bit}(Q_l^*)} - 2^{-2b_l^*} \right)$, measures how well the available bitwidths $\{\mathrm{bit}(Q) \mid Q \in \mathcal{Q}\}$ approximate the optimal bit allocation $\{b_l^*\}$.

To investigate how quantizer-set design affects each component of the gap, Table 6 reports the distortion and bit allocation gaps for LLaMA 3.1-8B under 2.5- and 3.25-bit constraints. Two key observations emerge:

- **Effect of quantizer quality.** VQ-2,3,4 and TCQ-2,3,4 show comparable bit allocation gaps, but TCQ-2,3,4 yields much smaller distortion gaps. Thus, their performance difference mainly stems from quantizer quality rather than bit allocation.
- **Effect of broader bitwidth support.** Comparing TCQ-2,3,4 to TCQ-ALL reveals a substantial reduction in bit allocation gap, demonstrating that richer fractional-bitwidth support enables more accurate bit allocation and a closer match to the theoretical ideal.

This analysis motivates the design of *Q-Palette*, which provides high-quality TCQ quantizers and broad fractional-bitwidth coverage to reduce both distortion and bit allocation gaps.

Analyzing these factors provides insight into improving practical quantizer designs and selecting more effective quantizer sets to reduce the quantization optimality gap. Motivated by this analysis, we specifically designed *Q-Palette* as a versatile set of fractional-bit quantizers, including TCQ, which closely approaches the theoretical distortion bound, and providing broad fractional bitwidth support.

## C  Additional details on quantizers in Q-Palette

In this section, we provide implementation details for each quantizer family in Q-Palette. For each quantizer, we describe: (i) codebook construction, (ii) dequantization, and (iii) quantization

procedures. The quantization step relies on quantizer-specific round-to-nearest (RTN) operators, with procedures differing based on data availability:

- **Data-free scenario:** We partition each weight matrix into scalar elements (NUQ) or vectors (VQ, TCQ). Each partition is independently quantized using the RTN operator, and their resulting binary representations are concatenated to form the final quantized weight representation.

- **Data-aware scenario:** We adopt a block LDLQ framework as introduced in previous methods [17, 49, 50]. Specifically, for each weight matrix, we perform quantization in a block-wise manner guided by Hessian approximations, with quantizer-specific block sizes: 1 for NUQ, 2 for VQ, and 16 for TCQ. This method sequentially processes weight rows from first to last, iteratively updating weights based on the Hessian and cumulative quantization errors, and quantizing each updated weight via the RTN operator. For detailed formulations and additional theoretical background, please refer to QUIP# and QTIP [49, 50].

## C.1 Non-uniform scalar quantization (NUQ)

**Codebook construction.** For NUQ at bitwidth $b$, we construct the LUT using `flash1dkmeans`, a fast 1D $k$-means algorithm [25], applied to $10^8$ randomly sampled standard Gaussian values. We set the number of clusters to $k = 2^b$, resulting in a LUT $\in \mathbb{R}^{2^b}$.

**Dequantization.** Given the LUT, a binary representation $\mathbf{r} \in \{0, 1\}^b$ is dequantized as:

$$\mathrm{dq}(\mathbf{r}; \mathrm{LUT}) := \mathrm{LUT}[\mathrm{int}(\mathbf{r})],$$

where $\mathrm{int}(\mathbf{r})$ converts the binary representation $\mathbf{r} \in \{0, 1\}^b$ to its corresponding integer index in the range $[0, 2^b - 1]$.

**Quantization.** NUQ's RTN operator $\mathrm{RTN} : \mathbb{R} \rightarrow \{0, 1\}^b$ maps a scalar input $v \in \mathbb{R}$ to the nearest LUT entry:

$$\mathrm{RTN}(v; \mathrm{LUT}) := \underset{\mathbf{r} \in \{0,1\}^b}{\mathrm{argmin}} |v - \mathrm{LUT}[\mathrm{int}(\mathbf{r})]|.$$

Quantization follows the general procedures described above for data-free and data-aware scenarios.

## C.2 Vector quantization (VQ)

**Codebook construction.** For VQ at bitwidth $b$, we construct the codebook using the scikit-learn implementation of the 2D $k$-means algorithm, which employs Lloyd's algorithm [41, 34]. We set the hyperparameters to `max_iter=300` and `tol=1e-6`, and apply the algorithm to random standard Gaussian samples, using $10^8$ samples for bitwidths $b \leq 5$ and $10^7$ samples for bitwidths $b > 5$. The number of clusters is set to $k = 2^{2b}$, resulting in a LUT $\in \mathbb{R}^{2^{2b} \times 2}$ consisting of $2^{2b}$ number of 2D vectors.

**Dequantization.** Given the LUT, a binary representation $\mathbf{r} \in \{0, 1\}^{2b}$ is dequantized similarly to NUQ, now mapping to a vector:

$$\mathrm{dq}(\mathbf{r}; \mathrm{LUT}) := \mathrm{LUT}[\mathrm{int}(\mathbf{r})] \in \mathbb{R}^2,$$

where $\mathrm{int}(\mathbf{r})$ converts the binary representation $\mathbf{r} \in \{0, 1\}^{2b}$ into its corresponding integer index in the range $[0, 2^{2b} - 1]$.

**Quantization.** The RTN operator specific to VQ, $\mathrm{RTN} : \mathbb{R}^2 \rightarrow \{0, 1\}^{2b}$, maps a 2D input vector $\mathbf{v} \in \mathbb{R}^2$ to the nearest LUT entry:

$$\mathrm{RTN}(\mathbf{v}; \mathrm{LUT}) := \underset{\mathbf{r} \in \{0,1\}^{2b}}{\mathrm{argmin}} \|\mathbf{v} - \mathrm{LUT}[\mathrm{int}(\mathbf{r})]\|_2.$$

Quantization then follows the general procedures described above for data-free and data-aware scenarios.

## C.3 Trellis-coded quantization (TCQ)

### C.3.1 Generic TCQ

**Codebook construction.** We follow the same protocol as QTIP [50], using scikit-learn's $k$-means implementation based on Lloyd's algorithm [41, 34]. Specifically, we cluster $2^{20}$ randomly sampled 2D standard Gaussian vectors (with appropriate scaling) into $2^{\texttt{tlut\_bits}}$ clusters, obtaining cluster centroids $\texttt{tlut} \in \mathbb{R}^{2^{\texttt{tlut\_bits}} \times 2}$. We then construct the final codebook $\text{LUT} \in \mathbb{R}^{2^{L} \times 2}$ using the *hybrid* codebook construction using the following `quantlut_sym` function from the QTIP codebase:

```
def quantlut_sym(tlut, L, tlut_bits):
    with torch.no_grad():
        lut = torch.arange(1 << L, device=tlut.device)
        lut = (lut + 1) * lut
        sflp = 1 - ((lut >> 15) & 1) * 2
        lut = (lut >> (16 - tlut_bits - 1)) & ((1 << tlut_bits) - 1)
    lut = tlut[lut]
    lut[:, 0] = lut[:, 0] * sflp
    return lut
```

Following QTIP, we set $L = 16$ for all TCQ quantizers. We set `tlut_bits` to 9 for bitwidths $b \le 4$, and to $10, 11$ for new fractional bitwidths $4.5, 5.0$, respectively.

**Dequantization.** We adopt the bitshift variant of TCQ with tail-biting from QTIP. Given a binary representation $\mathbf{r} \in \{0, 1\}^{sT/V}$, we define parameters explicitly as follows:

- $s$: shift size, set as $s = 2b$ for bitwidth $b$,
- $V$: vector size, fixed to $V = 2$,
- $L$: codebook length (or sliding window size), fixed to $L = 16$,
- $T$: trellis size, set as $T = 256$.

The dequantization then proceeds via sliding-window LUT indexing with tail-biting:

$$\text{dq}(\mathbf{r}; \text{LUT}) := \text{concat}_{i=0}^{T/V-1} \text{LUT}\left[\mathbf{r}[i \cdot s : i \cdot s + L]\right] \in \mathbb{R}^{T},$$

where indices exceeding the length of $\mathbf{r}$ wrap around due to tail-biting, resulting in $s/V$ bitwidth.

**Quantization.** Given a target vector $\mathbf{v} \in \mathbb{R}^{T}$ to quantize, we use the same RTN operator as detailed in QTIP, which leverages the Viterbi algorithm to find the optimal binary representation $\mathbf{r} \in \{0, 1\}^{sT/V}$ that is dequantized into the vector $\hat{\mathbf{v}} = \text{dq}(\mathbf{r}; \text{LUT})$ closest to the vector $\mathbf{v}$ [16, 50]. Quantization procedures follow the general data-free and data-aware frameworks described earlier.

### C.3.2 Half-TCQ

**Codebook construction.** For half-TCQ, which quantizes half of the weight using bitwidth $b$ and the other half using $b + 0.5$, we follow exactly the same codebook construction procedure described above for TCQ at bitwidth $b + 0.5$.

**Dequantization.** Dequantization separately processes two partitions of the weight matrix: the first half using binary representations of bitwidth $b$, and the second half using bitwidth $b + 0.5$. The resulting vectors from each half are then concatenated into a complete dequantized weight vector.

**Quantization.** We apply the RTN operator corresponding to TCQ-$b$ to the first half of the weights, and the RTN operator corresponding to TCQ-$(b + 0.5)$ to the second half. This procedure is consistently used in both the data-free and data-aware (block LDLQ) scenarios.

## D  Additional details and performance analysis of CUDA kernels

We implemented two types of CUDA kernels: (i) Tensor Core-based kernels and (ii) CUDA Core-based kernels. Here, we first detail our Tensor Core-based kernel implementations. Then, we describe

Table 7: Decoding-latency speedup of quantized LLaMA 3.1-8B models relative to the FP16 baseline on an RTX3090 GPU. 'TC' and 'CC' denote Tensor Core and CUDA Core kernels, respectively.

| | Decoding-latency speedup compared to FP16 (batch size = 1) | | | | | | | | | |
|---|---|---|---|---|---|---|---|---|---|---|
| Quantizer | 2.0 | 2.25 | 2.5 | 2.75 | 3.0 | 3.25 | 3.5 | 3.75 | 4.0 | 4.25 |
| NF w/ FLUTE [11, 20] | - | - | - | - | - | 1.63× | - | - | - | 1.63× |
| QTIP [50] | 2.17× | - | - | - | 2.02× | - | - | - | 1.92× | - |
| Ours-NUQ-TC | 2.82× | - | - | - | 2.53× | - | - | - | 2.28× | - |
| Ours-NUQ-CC | 3.07× | - | - | - | 2.75× | - | - | - | **2.38×** | - |
| Ours-VQ-TC | 2.83× | - | 2.54× | - | 2.54× | - | 2.10× | - | 2.28× | - |
| Ours-VQ-CC | **3.11×** | - | **2.94×** | - | 2.74× | - | 2.52× | - | 2.36× | - |
| Ours-TCQ-TC | 2.56× | **2.32×** | 2.29× | **2.24×** | 2.36× | **2.13×** | 2.14× | **2.16×** | 2.25× | **1.94×** |

| | Decoding latency speedup compared to FP16 (batch size = 8) | | | | | | | | | |
|---|---|---|---|---|---|---|---|---|---|---|
| Quantizer | 2.0 | 2.25 | 2.5 | 2.75 | 3.0 | 3.25 | 3.5 | 3.75 | 4.0 | 4.25 |
| NF w/ FLUTE [11, 20] | - | - | - | - | - | 1.55× | - | - | - | 1.55× |
| QTIP [50] | 0.43× | - | - | - | 0.39× | - | - | - | 0.36× | - |
| Ours-NUQ-TC | **2.20×** | - | - | - | **2.00×** | - | - | - | 1.85× | - |
| Ours-NUQ-CC | 1.75× | - | - | - | 1.70× | - | - | - | 1.49× | - |
| Ours-VQ-TC | **2.20×** | - | **1.99×** | - | 1.97× | - | 1.65× | - | 1.84× | - |
| Ours-VQ-CC | 1.88× | - | 1.81× | - | 1.78× | - | 1.69× | - | 1.69× | - |
| Ours-TCQ-TC | 2.04× | **1.89×** | 1.84× | **1.88×** | 1.93× | **1.79×** | 1.72× | **1.83×** | **1.89×** | **1.67×** |

our CUDA Core-based kernel implementations. Finally, we provide additional performance analysis on our kernels.

### D.1 Tensor Core-based kernel implementation

Our Tensor Core-based kernels support various quantization schemes (TCQ, NUQ, and VQ) and are implemented by extending the QTIP kernels, which originally supported TCQ at integer bitwidths (2, 3, 4 bits) [50]. Specifically, for TCQ, we introduce optimized support for fractional bitwidths at fine-grained intervals (*e.g.*, 1.5, 2.5, 3.5, 4.5, 5.0 bits) by carefully extending the warp-level `mma` instruction-based implementation provided by QTIP. Additionally, we adapted the QTIP's kernel design principles to implement efficient Tensor Core-based kernels for NUQ and VQ. These extensions involved non-trivial engineering efforts, particularly for precisely mapping quantized weights into Tensor Core `mma` instruction fragments. To further reduce overhead at larger batch sizes, we traverse each quantized weight exactly once, directly performing register-level dequantization upon loading without intermediate storage. Input activations are cached in shared memory to enable efficient reuse across multiple weight multiplications, substantially improving inference efficiency.

**Simplified Kernel Structure.** Below, we provide a brief kernel structure highlighting key functions, their purposes, and file locations:

```
// kernels/tcq-kernels/src/inference.cu
device void load_reg_cs<R>(compressed, idx, laneId, &regs) {
  // Maps quantized TCQ weights (bitwidth R/2) to mma fragments
  // Supports fractional bitwidths (1.5,2.0,2.5,...,4.5,5.0)
}

// kernels/vq-tensor-kernels/src/inference.cu
device void load_reg_cs<R, LUT_TYPE L>(compressed, idx, laneId, &regs) {
  if (L == LUT_TYPE::SQ_LUT) {
    // Maps quantized NUQ weights (bitwidth R) to mma fragments
  } else if (L == LUT_TYPE::VQ_LUT_2) {
    // Maps quantized VQ weights (bitwidth R/2) to mma fragments
  }
}

// General Tensor Core kernel structure
// - kernels/tcq-kernels/src/inference.cu: kernel_decompress_gemm
```

```
//      for TCQ fused kernel
// - kernels/tcq-kernels/src/inference.cu: kernel_decompress_gemm_combt
//      for TCQ-Half fused kernel
// - kernels/vq-tensor-kernels/src/inference.cu: kernel_decompress_gemm
//      for VQ and NUQ fused kernel
global void tensor_core_kernel(...) {
  // Manages LUT and inputs in shared memory
  // Manages quantized weights in registers
  // Maps quantized weights to Tensor Core mma fragments via 'load_reg_cs'
  // Uses Tensor Core mma instructions for matmul routine
  // Writes final results after reduction
}

// Dequantization-only kernels
// - kernels/tcq-kernels/src/inference.cu: kernel_decompress
//      for TCQ dequantization kernel
// - kernels/tcq-kernels/src/inference.cu: kernel_decompress_combt
//      for TCQ-Half dequantization kernel
// - kernels/vq-tensor-kernels/src/inference.cu: kernel_decompress
//      for VQ and NUQ dequantization kernel
global void kernel_decompress(...) {
  // Dequantizes weights independently (no matmul)
}
```

**Full Implementation.**   The complete Tensor Core-based kernel implementations are included in our public code release (directories `kernels/tcq-kernels` and `kernels/vq-tensor-kernels`).

### D.2   CUDA Core-based kernel implementation

Our CUDA Core-based kernels explicitly leverage CUDA Core instructions to support NUQ and VQ quantization schemes, extending the Any-Precision LLM kernels originally developed for NUQ [40]. Specifically, we replaced the original bit-plane encoding with simpler bit-packing encoding to streamline the dequantization procedure.

**Kernel Structure and Implementation**   Below, we provide a brief kernel structure highlighting key functions, their purposes, and file locations:

```
// - kernels/vq-cuda-kernels/src/gemm_routines.cu
device void vq_pack_dequant_routine<nbits, vec_sz>(Bcode, B_row, shC) {
  // Unpack quantized VQ weights 'Bcode' (bitwidth nbits/vec_sz)  \
  //      to the half2 array 'B_row' using the lookup-table 'shC' \
  //      (e.g., 1.5-bit quantization: nbits=3, vec_sz=2)
}

// - kernels/sq-cuda-kernels/gemm_routines.cu
device void pack_dequant<nbits>(Bcode_row, B_row, shC) {
  // Unpack quantized NUQ weights 'Bcode' (bitwidth nbits) \
  //      to the half2 array 'B_row' using the lookup-table 'shC'
}

// General CUDA Core kernel structure
// - kernels/sq-cuda-kernels/gemm_routines.cu: sq_gemm_fp16
//      for NUQ fused kernel
// - kernels/vq-cuda-kernels/src/gemm_routines.cu: vq_pack_gemm_fp16
//      for VQ fused kernel
global void cuda_core_kernel(...) {
  // Manages LUT in shared memory
```

```
  // Manages quantized weights in registers
  // Unpacks quantized weights to half2 array via corresponding
  //     pack_dequant_routine
  // Uses CUDA Core half-precision FMA (hfma2) instructions for matmul
  // Writes final results after reduction
}

// Dequantization-only kernels
// - kernels/sq-cuda-kernels/gemm_routines.cu: pack_dequant_kbit_store
//     for NUQ dequantization kernel
// - kernels/vq-cuda-kernels/src/gemm_routines.cu: \
//                                    vq_pack_dequant_kbit_store
//     for VQ dequantization kernel
global void kernel_decompress(...) {
  // Dequantizes weights independently (no matmul)
}
```

**Full Implementation.** The complete CUDA Core-based kernel implementation is included in our public code release (directories `kernels/sq-cuda-kernels` and `kernels/vq-cuda-kernels`).

### D.3 Additional performance analysis

This section complements Table 2 in the main paper by providing additional performance analysis on a different hardware configuration (with RTX3090 GPU). Table 7 summarizes the decoding-latency speedup of quantized LLaMA 3.1-8B models relative to the FP16 baseline on an RTX3090 GPU, complementing Table 2 in the main paper, which reports results on an RTX4090 GPU. Our quantizers consistently outperform baseline methods across both evaluated batch sizes (1 and 8).

Notably, for batch size 1 on RTX3090, the latency speedup gap between our CUDA Core-based ('CC') and Tensor Core-based ('TC') kernels is larger than observed on RTX4090 (Table 2). This difference highlights significant hardware dependencies in kernel performance, justifying the importance of providing multiple kernel implementations. Such flexibility enables optimal kernel selection tailored to specific hardware platforms and workload requirements.

## E    Implementation details of mixed-scheme quantization

### E.1    Loss term computation

**Data-free loss term.** In data-free scenarios, we estimate the loss term via the linearity theorem [35], *i.e.*, $\ell_{lq} = a_l \cdot \mathrm{err}(Q_q; W_l)$. Specifically, we approximate the quantization error $\mathrm{err}(Q_q; W_l)$ using pre-computed distortion values obtained by quantizing random standard Gaussian matrices, thereby avoiding explicit quantization for each weight matrix $W_l$. To determine the sensitivity coefficient $a_l$, we adopt the procedure introduced in HIGGS [35]. First, we randomly generate 128K tokens from the given LLM. Then, for each layer $l$, we inject random Gaussian noise scaled to specific norms $n_{li} = \frac{\sqrt{i}}{16}$ for $1 \leq i \leq 16$ and measure the resulting increase in the KL-divergence loss computed over these 128K tokens:

$$\Delta \mathcal{L}_{li} = \mathcal{L}\left(\{W_{l'} + \delta_{l'l} \cdot n_{li} \cdot \|W_l\|_2 \cdot \epsilon_l / \|\epsilon_l\|_2\}_{l'=1}^L\right) - \mathcal{L}\left(\{W_{l'}\}_{l'=1}^L\right), \quad \epsilon_l \sim \mathcal{N}(0, I),$$

where $\delta_{l'l}$ is the Kronecker delta (1 if $l' = l$, else 0), ensuring noise is injected exclusively into layer $l$, and $\epsilon_l$ is a standard Gaussian noise matrix matching the dimensions of layer $l$. Due to the linearity theorem, the increase in loss approximately follows the linear relation $\Delta \mathcal{L}_{li} \simeq n_{li}^2 a_l$, enabling us to estimate $a_l$ by linearly fitting the data points $(n_{li}^2, \Delta \mathcal{L}_{li})$. This procedure requires $16 \times L$ computations of the KL-divergence loss, where $L$ is the total number of layers, but it can be performed in an embarrassingly parallel manner. Furthermore, the computed sensitivity coefficients $a_l$ can be reused for all data-free MSQ scenarios, incurring only a one-time computational cost.

**Data-aware loss terms.** For data-aware scenarios, we employ two different types of loss terms: *linearity* and *actual*.

- **Linearity-based loss term.** Similar to the data-free scenario, we utilize the linearity theorem-based approximation. However, we replace the KL-divergence loss computed over randomly generated 128K tokens with the perplexity loss computed over 1M tokens from the RedPajama dataset [52].

- **Actual loss term.** Here, we explicitly calculate the actual perplexity increase caused by quantization. Specifically, we use 256K tokens from the RedPajama dataset and compute the loss term as follows:

$$\ell_{lq} := \mathcal{L}(\{Q_{\text{default}}^{l'}(W_{l'})\}, \text{replace } l\text{-th layer with } Q_q(W_l)) - \mathcal{L}(\{Q_{\text{default}}^{l'}(W_{l'})\}),$$

where $Q_{\text{default}}^{l}$ denotes the default quantizer for layer $l$ (*e.g.*, we use TCQ-2 for 2-bit quantization, TCQ-3 for 3-bit quantization). This explicitly measures the actual empirical perplexity increase caused by quantize layer $l$ with a quantizer $Q_q$.

For the data-aware experiments in Table 4 of the main paper, we exclusively employ the 'actual' loss term. Additionally, in the ablation study provided in Table 9 (see Appendix G), we explicitly compare the performance using the two types of loss terms, 'actual' and 'linearity'.

## E.2 Latency profiling

For latency profiling, we measure the execution time of the CUDA kernels corresponding to each quantizer. Since computations such as normalization layers, rotations, and self-attention operations remain identical across quantizers, we specifically profile the latency of each fused dequantization and matrix multiplication kernel. To accurately estimate the overall inference latency, we first measure the end-to-end inference latency of several randomly selected quantization configurations using `torch.compile`. We then subtract the sum of kernel latencies to estimate the latency overhead caused by common computations (*e.g.*, normalization layers), uniformly distributing this overhead across all quantizer profiles.

When considering fused kernels, we separately measure kernel latencies corresponding to each fusion pattern. To account for latency overhead variations due to different fusion patterns, we adjust this overhead bias accordingly.

## E.3 Optimizer

To solve the mixed-scheme quantization optimization problem, we employ Google's OR-Tools optimization suite [42], specifically utilizing the SCIP solver with a time limit of 60 seconds.

# F Settings for Figure 1

In Figure 1, we present qualitative comparisons among quantization frameworks based on Q-Palette and the NormalFloat baseline with FLUTE kernels [11, 20], evaluated on the LLaMA 3.1-8B model using an RTX4090 GPU at a batch size of 1. Specifically, we validate WikiText2 perplexity at a sequence length of 8192 and measure the inference speedup compared to the FP16 baseline under the following detailed settings:

**NormalFloat (Baseline).** For NormalFloat, we employ FLUTE with a codebook size of $2^3$ and a group size of 64, resulting in an average bitwidth of 3.25. We utilize the FLUTE kernels released in the FLUTE codebase [11, 20], optimized for inference on an RTX4090 GPU, to measure inference latency and compute the speedup.

**Single-scheme quantization with TCQ-3.25 (Ours).** We apply data-free quantization uniformly to all layers of the LLaMA 3.1-8B model using our TCQ-3.25 quantizer from Q-Palette. This corresponds to the half-TCQ scheme which quantizes half of the weight matrix at bitwidth 3.0 and the other half at 3.5.

**MSQ with Q-Palette (Ours).** We leverage the full set of quantizers available in Q-Palette as our search space. For NUQ and VQ quantizers, both Tensor-Core and CUDA-Core kernel implementations are considered during optimization. Sensitivity coefficients $a_l$ are computed following the

HIGGS protocol [35] as detailed in Appendix E, and we utilize pre-computed distortion values as explained in Section 4.1 of the main paper. Given these sensitivity coefficients, distortion values, and pre-profiled latency measurements, we solve the latency-constrained MSQ optimization (Equation (3) in the main paper) to identify Pareto-optimal quantizer selections under various latency constraints. From this resulting accuracy-latency trade-off curve, we select the configuration that clearly improves both latency and perplexity over the TCQ-3.25 baseline.

**Fusion-aware MSQ with Q-Palette (Ours).** We further incorporate layer fusion into our MSQ formulation by additionally profiling latency measurements for fused linear-layer combinations and solving the fusion-aware optimization (Equation (4) in the main paper). This approach explicitly captures the latency reductions achievable via layer fusion, enabling joint optimization of quantization schemes and layer fusion decisions. We select a quantization configuration that clearly improves both inference speed and perplexity compared to the MSQ baseline without layer fusion.

The detailed quantization configurations correspond to these scenarios are visualized in Figure 1.

# G Experimental settings and additional results

## G.1 Experimental settings

### G.1.1 Evaluation metric details

For evaluating language modeling performance, we measure perplexity on the WikiText2 dataset [37], using sequence lengths of 4096 tokens for LLaMA 2 models and 8192 tokens for LLaMA 3 models. Additionally, we report zero-shot accuracy on five downstream tasks: ARC-easy, ARC-challenge, HellaSwag, PiQA, and WinoGrande [37, 8, 56, 3, 44]. Zero-shot evaluations are conducted using the `lm_eval` library (version 0.4.4).

### G.1.2 Device details

**RTX 4090 GPU experiments.** We conduct our RTX 4090 GPU experiments using a cloud environment provided by RunPod, with the following hardware and software specifications:

- **GPU:** NVIDIA RTX 4090
- **CPU:** AMD EPYC 7B13 64-Core Processor
- **OS:** Ubuntu 22.04.5
- **CUDA Version:** 12.4

**RTX 3090 GPU experiments.** We conduct our RTX 3090 GPU experiments using our local machine, detailed as follows:

- **GPU:** NVIDIA RTX 3090
- **CPU:** AMD EPYC 7402 24-Core Processor
- **OS:** Ubuntu 22.04.1
- **CUDA Version:** 12.4

### G.1.3 Baseline configurations

**HQQ [2].** According to the official documentation, inference acceleration kernels (*e.g.*, Gemlite) are supported only for configurations with `axis=1`. Thus, we use the following configurations:

- 4.25-bit: `nbits=4, group_size=64, axis=1` (Gemlite kernel),
- 4.02-bit: `nbits=4, group_size=1024, axis=1` (Gemlite kernel),
- 3.25-bit: `nbits=3, group_size=64, axis=1` (FLUTE kernel).

For 4-bit instances, we utilize Gemlite kernels following best practices from the HQQ documentation [22, 2]; for the 3-bit instance, we report inference time using the FLUTE kernel [20].

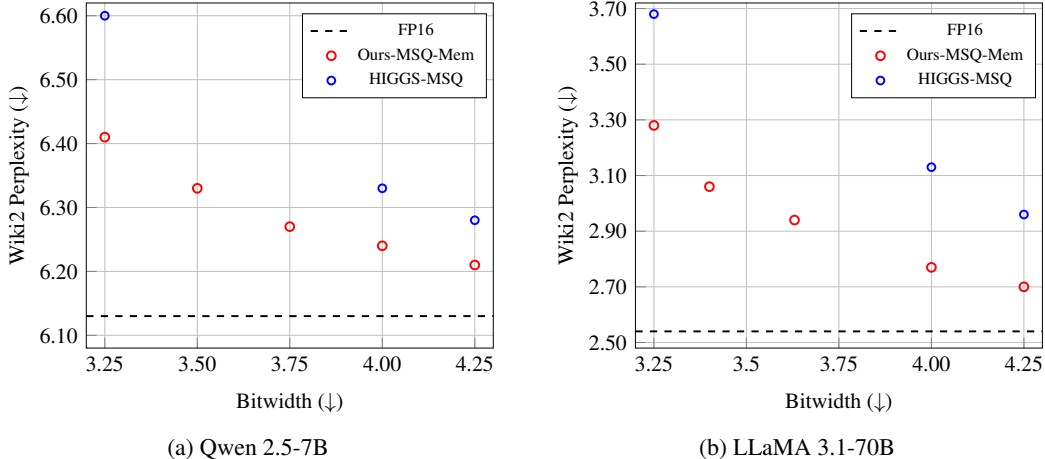

|               |                 |
|---------------|-----------------|
| (a) Qwen 2.5-7B | (b) LLaMA 3.1-70B |

Figure 5: Mixed-scheme quantization results on Qwen 2.5-7B and LLaMA 3.1-70B models. To accommodate the broader sensitivity range in LLaMA 3.1-70B, we extended the quantizer set to include higher-bitwidth options (NUQ 7/8 bits and VQ 5.5/6 bits), in addition to the TCQ quantizers.

**NormalFloat [11]** We utilize FLUTE's NormalFloat implementation with configurations similar to HQQ [20, 2]:

- 4.25-bit: `nbits=4, group_size=64`,
- 4.02-bit: `nbits=4, group_size=1024`,
- 3.25-bit: `nbits=3, group_size=64`.

Since the publicly available optimized FLUTE kernel does not support a group size of 1024, we only report inference latency results for the 4.25-bit and 3.25-bit configurations.

**QTIP [50].** For data-free QTIP, we approximate the Hessian as the identity matrix and follow the same algorithmic implementation as the original data-aware QTIP. For data-aware QTIP, we use the publicly available Hessian approximation from the `relax-ml` HuggingFace repository, computed using $6144 \times 4096$ tokens.

**HIGGS [35].** As the implementation of HIGGS is not publicly available, we directly report results from their paper. Specifically, for the mixed-scheme baseline, HIGGS provides only a single result for each bitwidth, which we directly use in our comparison. For the single-scheme VQ baseline, HIGGS reports multiple configurations for each bitwidth; we select the configuration achieving the lowest WikiText2 perplexity. Although these single-scheme VQ configurations may not be efficiently realizable in practice due to non-power-of-two codebook sizes, we include them for completeness and comparison purposes.

### G.2 Additional results

#### G.2.1 Memory-constrained mixed scheme quantization results on additional models

To demonstrate the generality of our method, we applied our MSQ method to Qwen 2.5-7B (non-LLaMA) and LLaMA 3.1-70B (large-scale), comparing against HIGGS-MSQ under various bitwidth constraints [55, 18, 35]. Here, for LLaMA 3.1-70B, due to its substantially larger model size, we used 64K tokens (*i.e.*, half of the default setting explained in Appendix E) to estimate the sensitivity coefficients while keeping the rest of the quantization pipeline unchanged.

As shown in Figure 5, our method consistently outperforms HIGGS-MSQ under the same bitwidth constraints (3.25, 4.00, 4.25) on both models. Additionally, our method achieves comparable or better perplexity at lower bitwidths compared to HIGGS-MSQ. For Qwen 2.5-7B, our 3.5-bit model matches the performance of HIGGS-MSQ at 4.00 bits, and our 3.75-bit result slightly improves upon the HIGGS-MSQ result at 4.25 bits, yielding up to 12.5% memory savings. A similar trend is observed

Table 8: Ablation study of layer fusion and CUDA-Core kernel usage on inference throughput and WikiText2 perplexity (LLaMA 3.1-8B, batch size=1, RTX4090). 'TC' and 'CC' denote Tensor Core and CUDA Core kernels, respectively.

| Method | Throughput (Toks/s) ($\uparrow$) | Wiki2 ($\downarrow$) |
|---|---|---|
| FP16 | 62 | 5.61 |
| Ours-VQ-2 (single scheme) | 231 | 5905.08 |
| Ours-MSQ-Lat (No Fusion, TC Only) | 228 | 119.72 |
| Ours-MSQ-Lat (Fusion-aware, TC only) | 232 | 8.93 |
| Ours-MSQ-Lat (Fusion-aware, TC and CC) | 231 | **8.71** |
| Ours-TCQ-2 (single scheme) | 223 | 37.95 |
| Ours-MSQ-Lat (No Fusion, TC Only) | 223 | 20.33 |
| Ours-MSQ-Lat (Fusion-aware, TC only) | 224 | 7.79 |
| Ours-MSQ-Lat (Fusion-aware, TC and CC) | 223 | **7.69** |
| Ours-TCQ-3 (single scheme) | 185 | 6.78 |
| Ours-MSQ-Lat (No Fusion, TC Only) | 187 | 6.47 |
| Ours-MSQ-Lat (Fusion-aware, TC only) | 186 | 6.06 |
| Ours-MSQ-Lat (Fusion-aware, TC and CC) | 185 | **6.03** |

Table 9: Ablation comparing 'linearity' vs. 'actual' loss terms in data-aware MSQ (LLaMA 2-7B, RTX4090 GPU)

| | LLaMA 2 7B | | | | |
|---|---|---|---|---|---|
| | | | | Throughput (Toks/s) | |
| Method | Bits | Wiki2 ($\downarrow$) | Acc ($\uparrow$) | $B = 1$ | $B = 8$ |
| FP16 | 16.00 | 5.12 | 64.9 | 71 | 527 |
| QTIP | 2.00 | 6.84 | 58.9 | 209 | 386 |
| Ours-MSQ-Mem (linearity) | 2.00 | 6.69 | **60.3** | 270 | **1690** |
| Ours-MSQ-Mem (actual) | 2.00 | **6.47** | **60.3** | **272** | 1684 |
| QTIP | 3.00 | 5.39 | 63.3 | 184 | 304 |
| Ours-MSQ-Mem (linearity) | 3.00 | 5.38 | **64.3** | **225** | **1501** |
| Ours-MSQ-Mem (actual) | 3.00 | **5.34** | 63.9 | 224 | 1489 |

on LLaMA 3.1-70B, where our 3.40-bit and 3.63-bit results slightly outperform HIGGS-MSQ at 4.00 and 4.25 bits, respectively, resulting in up to 15% memory savings at better perplexity. These results demonstrate the broad applicability of our MSQ method.

### G.2.2 Ablation on layer fusion and CUDA-Core kernel integration

Table 8 presents an ablation study evaluating the impact of layer fusion and the integration of CUDA-Core kernels on inference throughput and WikiText2 perplexity for the quantized LLaMA 3.1-8B model. We compare single-scheme baselines (VQ-2, TCQ-2, and TCQ-3) in Q-Palette against several MSQ variants: MSQ without layer fusion (Tensor Core kernels only), fusion-aware MSQ using only Tensor Core kernels, and fusion-aware MSQ combining both Tensor Core and CUDA Core kernels. Results demonstrate significant improvements in WikiText2 perplexity when applying fusion-aware MSQ compared to single-scheme quantization and MSQ without fusion, highlighting the effectiveness of jointly optimizing quantization schemes and layer fusion. For example, MSQ without fusion achieves 20.33 perplexity at 223 tokens/sec, while our fusion-aware MSQ achieves a significantly reduced perplexity of **7.79** at 224 tokens/sec. Additionally, integrating CUDA Core kernels alongside Tensor Core kernels provides further performance improvement.

Table 10: Per-group quantization results on LLaMA 3.1-8B (group size = 64) without IP. For MSQ, we used a pool of per-group NUQ quantizers ranging from 2 to 8 bits with group size 64. All results are reported without IP.

| Method | Bitwidth | Bit allocation strategy | Wiki2 ($\downarrow$) |
|---|---|---|---|
| G64-NUQ-3 | 3.25 | Uniform bitwidth | 8063.91 |
| G64-MSQ-Mem | 3.25 | MSQ with Gaussian-assumed error | 7.34 |
| G64-MSQ-Mem | 3.25 | MSQ with true quantization error | **7.33** |
| G64-NUQ-4 | 4.25 | Uniform bitwidth | 6.10 |
| G64-MSQ-Mem | 4.25 | MSQ with Gaussian-assumed error | **5.89** |
| G64-MSQ-Mem | 4.25 | MSQ with true quantization error | 5.90 |

### G.2.3 Effect of loss-term choice in data-aware MSQ

Table 9 provides an ablation study comparing two different loss-term definitions ('linearity' vs. 'actual') used in data-aware MSQ quantization for LLaMA 2-7B. The 'linearity' loss term efficiently approximates the increase in perplexity loss via sensitivity coefficients measured using the linearity theorem, enabling reuse across multiple quantization configurations, similar to the data-free scenario (see Appendix E). In contrast, the 'actual' loss term explicitly computes the empirical (validation) perplexity increase caused by quantization. Our results demonstrate that using the computationally efficient 'linearity' loss term achieves comparable zero-shot accuracy improvements to those obtained with the 'actual' loss term, indicating that the simpler and reusable linearity-based approach is also effective in practice. Additionally, both loss-term approaches achieve similar inference throughput, reinforcing the practicality of the computationally efficient 'linearity' loss term.

### G.2.4 Applicability of MSQ with linearity-theorem surrogate without incoherence processing

A natural question is whether the MSQ framework based on the linearity-theorem surrogate [35] remains applicable when incoherence processing (IP) is not available due to the hardware constraints. This surrogate objective requires sensitivity coefficients and per-layer quantization errors; with IP, weights are nearly Gaussian, allowing these errors to be precomputed from random Gaussian matrices as explained in Appendix E.

To examine the no-IP case, where per-group quantization is typically adopted to handle weight outliers, we disable IP and quantize on a per-group basis (group size = 64) using fixed Gaussian-trained codebooks. We compare three bit allocation strategies: 1) uniform bitwidth, 2) MSQ with *Gaussian-assumed error*, which relies on cached Gaussian distortion estimates for $\mathrm{err}(Q_q; W_l)$ and solves Equation (1), and 3) MSQ with *true quantization error*, which measures layerwise distortion $\mathrm{err}(Q_q; W_l)$ directly and solves Equation (1).

As shown in Table 10, MSQ significantly outperforms uniform bitwidth even in the no-IP case. Moreover, the Gaussian-assumed error achieves perplexity almost identical to that from the true quantization error (*e.g.*, 7.34 vs. 7.33 at 3.25 bits), providing preliminary evidence that cached Gaussian-based error estimates may remain reliable in the no-IP case.

## H  Limitations and future work

We introduce Q-Palette, a comprehensive suite of quantizers spanning a wide range of trade-offs across memory footprint, inference latency, and quantization error, offering versatile options suitable for diverse deployment scenarios. To demonstrate its effectiveness, we integrate Q-Palette into an MSQ framework and validate its ability to achieve improved performance-efficiency trade-offs under PTQ settings. However, our framework is designed around one-shot MSQ objectives, which rely on layer-wise second-order approximations of end-to-end loss and are primarily applicable to scenarios that do not involve retraining [35]. While Q-Palette can also serve as a building block for retraining-based quantization workflows such as quantization-aware training, which may be preferable in cases where larger computational budgets and data are available, we have not evaluated its effectiveness in that setting, as our primary focus is on data-free or calibration-light PTQ. Extending Q-Palette to retraining-based quantization workflows thus remains a promising direction for future work.

Another limitation lies in the cost of computing the sensitivity coefficients. Currently, evaluating these coefficients requires $O(L)$ computations of the KL-divergence loss in data-free setting as explained in Appendix E, which can become a bottleneck as the model size grows. Although this computation can be performed in an embarrassingly parallel manner and the resulting coefficients can be reused across all MSQ runs, thus representing a one-time cost, the overhead may still be non-negligible when the set of target memory or latency is fixed and reuse is limited. Developing methods to further reduce this cost is therefore an interesting direction for future research.

One promising direction is the extension of Q-Palette to weight-activation quantization. In this work we focus on weight-only PTQ, which is particularly effective in memory-bound inference settings with small batch sizes, such as on laptops or mobile devices where memory bandwidth, rather than compute, is the primary bottleneck. However, on some hardware accelerators, such as the Qualcomm Hexagon NPU, which natively support only integer (e.g., INT8) GEMM, activation quantization is essential for exploiting their full performance. Thus, extending Q-Palette to support weight-activation quantization is a natural direction for broader deployment. One potential approach is a two-stage scheme: (1) first quantize weights to INT8 using uniform W8A8 quantizers for hardware compatibility; and (2) then apply a secondary compression step that further quantizes the INT8 weights into $x$-bit representations using a variant of Q-Palette quantizers whose codebooks are constrained to the INT8 grid. During inference, the compressed weights are dequantized back to INT8 and then processed using integer GEMM with INT8-quantized activations, enabling compatibility with INT8-only hardware while reducing memory usage. A similar idea was introduced in Q-Serve, which quantizes weights in two stages, first to symmetric INT8 grid and then to asymmetric INT4 [31]. We consider exploring such extensions an interesting direction for future work.

