# OpenReview forum: "Q-Palette: Fractional-Bit Quantizers Toward Optimal Bit Allocation for Efficient LLM Deployment"
_NeurIPS.cc/2025/Conference — NeurIPS 2025 poster_

### Official Review · Reviewer_4xNx · 2025-06-09

**Clarity:** 3
**Significance:** 3
**Originality:** 2
**Rating:** 4
**Confidence:** 5

**Summary:**

This paper aims to develop a finegrained fractional bit quantizer to approach the optimal bit allocation rate for Gaussianized weights after rotation-based transformation. The paper proposes a collection of fractional bit quantizers and corresponding CUDA kernels to achieve optimal compression rate and speedup.

**Questions:**

Please explain the difference between the proposed quantizer and the TCQ quantizer cited in the paper.

**Ethical Concerns:**

["NO or VERY MINOR ethics concerns only"]

**Final Justification:**

After checking the rebuttal and the author's reply, I'm confident in my judgement that the paper is solid. Though the novelty on the algorithm side may be limited, the practical contribution is adequate and should be encouraged.

**Limitations:**

Yes

**Paper Formatting Concerns:**

No concerns

**Quality:**

3

**Strengths And Weaknesses:**

## Strength
1. This paper pays special attention to the quantizer design for the post-rotation weight distribution, which is an area largely ignored by previous rotation-based quantizer design.
2. The paper makes solid emperical contribution by exploring the fractional-bit quantizers and the implementation of CUDA kernel, achieving both improved performance and real speedup
3. The paper is well-motivated, clearly-written, and easy to follow.

## Weakness
1. Novelty-wise, the fractional precision quantization and the kernel implementation proposed in the paper appears to have been based on existing work, as cited in the paper. The contribution of the paper is mainly implementing these techniques on rotational-quantized LLMs
2. The conclusion that fractional-bit quantization leads to better performance-efficiency tradeoff is not suprising. Though the paper brings practical advances on the model compression frontier, the research insight is less significant.

---

> ### Author Rebuttal · Authors · 2025-07-31
>
> Thank you for your thoughtful and constructive review. We appreciate your positive assessment that the paper is “well-motivated, clearly-written, and easy to follow,” and that it makes a “solid empirical contribution” by exploring fractional-bit quantizers and implementing CUDA kernels that achieve both improved performance and real speedup.
>
> ---
>
> **Regarding our contributions and novelty (W1 & W2).**
>
> As the reviewer correctly noted, one of the key contributions of this work lies in the practical realization of fractional-bit quantizers through efficient CUDA kernel implementations. Supporting a wider range of bitwidths and larger batch sizes required substantial engineering effort, including the precise mapping of quantized weights to `mma` instruction fragments for each quantizer configuration. These implementations make Q-palette quantizers practically deployable and enable strong empirical results across various model sizes and compression levels.
>
> Beyond implementation, our work offers contributions and novel ideas in several other aspects, summarized below:
>
> * First, we introduce **fusion-aware MSQ**, the first framework to jointly optimize operator fusion decision and quantizer selection under resource constraints. Within each Transformer block, certain linear layers, such as {query, key, value}, share the same input and can be fused into a single matrix multiplication. For example, instead of separately computing $ \\mathbf{x}W_q$, $ \\mathbf{x}W_k$, and $ \\mathbf{x}W_v$ for an input $\\mathbf{x}$, we can concatenate the weight matrices and compute $\\mathbf{x}(W_q \oplus W_k \oplus W_v) $, followed by splitting the output. This fusion technique is often used to improve inference speed by reducing kernel launches and memory accesses. Our method integrates operator fusion directly into the MSQ framework, formulating a unified ILP. This leads to substantial improvements in the accuracy-latency trade-off. For example, on LLaMA 3.1-8B (batch size 1, RTX 4090), standard MSQ achieves 20.33 perplexity at 223 toks/sec, while our fusion-aware MSQ reduces perplexity to **7.79** at a similar throughput of **224** toks/sec (Appendix G, Table 7). Qualitative results are shown in Figure 1.
>
> * In addition, Section 3.1 provides a **theoretical motivation** for the design of Q-Palette. Theorem 3.1 and the analysis in Appendix B show that to improve MSQ performance, it is important to use practical quantizers that achieve distortion close to the theoretical optimum for a given bitwidth, and are available at fine-grained fractional bitwidths. This insight motivates the design of Q-Palette and offers guidance for future quantizer development tailored to optimal bit allocation.
>
> * We also introduce a novel quantization scheme, **half-TCQ**, which splits a weight matrix and applies different TCQ bitwidths to each half (*e.g.*, 2.5 and 3.0 bits), achieving a 2.75-bit representation. This enables finer-grained bitwidth control beyond 0.5-bit intervals. To support efficient deployment, we implement CUDA kernels that perform dequantization and matrix multiplication for a half-TCQ layer in a single kernel call.
>
>
> We will revise the introduction in the camera-ready version to more clearly reflect these contributions.
>
> ---
>
> **Q1. Please explain the difference between the proposed quantizer and the TCQ quantizer cited in the paper.**
>
> While our TCQ quantizers are based on the bitshift variant of TCQ introduced in QTIP, our TCQ quantizers introduce several key enhancements that significantly improve practicality.
>
> First, while QTIP supports only a limited set of integer bitwidths (2, 3, 4 bits), our implementation extends support to a broader range of fractional bitwidths (1.5, 2.0, 2.5, 3.0, 3.5, 4.0, 4.5, 5.0 bits). In addition, we introduce half-TCQ, a novel scheme that enables intermediate bitwidths such as 1.75, 2.25, 2.75, 3.25, 3.75, 4.25, 4.75 by applying different bitwidths to each half of the weight matrix.
>
> Moreover, our kernels support batch sizes up to 8, whereas QTIP was restricted to batch size 1, substantially improving deployment flexibility.
>
> Finally, we reduce runtime overhead by rotating weights only along the input dimension and reusing rotation matrices across layers that share inputs, effectively reducing the number of online rotations (Section 3.3).

---

> > ### Comment · Reviewer_4xNx · 2025-08-03
> >
> > I would like to thank the author for the response. I'm generally satisfied with the overall contribution of the paper and I'm more confident with my assessment after reading the author response and other reviews.

---

### Official Review · Reviewer_hvcr · 2025-06-21

**Clarity:** 3
**Significance:** 3
**Originality:** 3
**Rating:** 4
**Confidence:** 4

**Summary:**

This paper has two main parts: a bit allocation algorithm that chooses how many bits to compress each linear layer with, and a series for scalar, vector, and trellis coded quantization that enable the bit allocation algorithm to be useful in practice. The bit allocation algorithm builds upon the "linearity theorem" in HIGGS by formulating bit allocation as a multi choice knapsack problem. The kernels build upon existing quantizers in the literature, with the main contribution being a set of fast trellis coded kernels that support a wider range of bitrates and decoding batch sizes than the existing QTIP kernels while also being faster. The empirical results in this paper show that Q-Palette's bit allocation algorithm achieves better compression-distortion tradeoffs than uniform bitrate methods and that Q-Palette's kernels are more flexible and faster than existing kernels.

**Questions:**

See above

**Ethical Concerns:**

["NO or VERY MINOR ethics concerns only"]

**Final Justification:**

(copy of my final response to the authors). Thanks for your follow up experiments. I think they confirm my initial thoughts about the paper, which is that the bit allocation algorithm and better tcq kernels are somewhat orthgonal to each other. The bit allocation theorem could also be improved due to the possibility of just breaking the model in half (which apparently happens empirically without a bit floor). However, I think both components of the paper are useful and practical contributions that are not just incremental. I think this paper should be accepted, and if we were on the old scale I would give it a weak accept, which seems to be a 4.5 on this scale. I will keep my score at 4 for now since IMO a 5 would be a 7-8 on the old scale, and I don't think this paper is at the level of 7-8.

**Limitations:**

Yes

**Quality:**

3

**Strengths And Weaknesses:**

Strengths:
- This paper has two key components: a bit allocation algorithm and a set of kernels. Based on the empirical results, the kernels are faster and more flexible than the existing QTIP kernels without sacrificing quality. This alone is an important contribution, since a major hurdle to adopting advanced quantizers is good kernel support. The bit allocation algorithm works well in practice and has some basis in theory, although I have some questions (see below).
- The bit allocation algorithm works with data-free quantization algorithms.
- The paper is well written and easy to follow.
- Although this is a more "minor" part of the paper, the authors introduce a way to perform fractional trellis coded quantization within a single weight matrix. To the best of my knowledge, this is new and nice to see.

Weaknesses/Questions:
- The bit allocation algorithm assumes that the input weight distribution has been whitened to be ~ iid Gaussian with incoherence processing. It isn't always possible to apply IP due to hardware or latency constraints. Does the bit allocation algorithm still work when the weight distribution isn't iid Gaussian? That is, if you produce a set of bitrates with a Gaussian assumption and then quantize a model with these bitrates but without IP, do you still get an improvement over the uniform bitrate case?
- Theorem 3.1 seems suspicious to me. max(0, bitrate) implies that if the computed bitrate is negative, this linear layer should be pruned from the model since it gets 0 bits. Surely this isn't correct?
- The paper says that the bit allocation algorithm works better with a better gaussian quantizer (ie closer to the rate distortion limit). The main experiments mostly use the QP trellis quantizers. Do you have experiments that show that the bit allocation algorithm performs worse when used in the VQ or SQ case?

---

> ### Author Rebuttal · Authors · 2025-07-31
>
> Thank you for your thoughtful and encouraging review. We sincerely appreciate your recognition that the paper is "well written and easy to follow," that our fractional TCQ variant is "new and nice to see," and that the proposed bit allocation algorithm "works well in practice and has some basis in theory." Your insightful questions helped us further validate and clarify key aspects of our work.
>
> ---
>
>
> **W1. Does the bit allocation algorithm still work without incoherence processing (IP)?**
>
> We appreciate this important question regarding the robustness of our bit allocation strategy when the Gaussianity assumption is violated.
>
> To assess this, we compare single-scheme quantization and mixed-scheme quantization (MSQ) on LLaMA 3.1-8B using non-uniform scalar quantizers (NUQ), in a setting where IP is disabled.
>
> *Table C. Performance comparison at 3-bit quantization with and without IP on LLaMA 3.1-8B.*
>
> | Method       | IP | Bit allocation strategy          | Perplexity ↓ |
> | ------------ | -- | ---------------------------------- | ------------ |
> | Ours-NUQ-3   | ✗  | Uniform bitwidth                             | 36.86        |
> | Ours-MSQ-Mem | ✗  | Bit allocation by Gaussian-assumed error             | 11.73        |
> | Ours-MSQ-Mem | ✗  | Bit allocation by true quantization error                 | 10.76        |
> | Ours-MSQ-Mem | ✓  | Bit allocation by Gaussian-assumed error             | 6.84     |
>
> *Table D. Performance comparison at 4-bit quantization with and without IP on LLaMA 3.1-8B.*
>
> | Method       | IP | Bit allocation strategy          | Perplexity ↓ |
> | ------------ | -- | ---------------------------------- | ------------ |
> | Ours-NUQ-4   | ✗  | Uniform bitwidth           | 6.21         |
> | Ours-MSQ-Mem | ✗  | Bit allocation by Gaussian-assumed error             | 6.17         |
> | Ours-MSQ-Mem | ✗  | Bit allocation by true quantization error | 6.13         |
> | Ours-MSQ-Mem | ✓  | Bit allocation by Gaussian-assumed error             | 5.92     |
>
> Tables C and D show the quantization result for various bit allocation strategies without IP. Even without IP, our MSQ method significantly outperforms uniform bitwidth setting. For example, at 3 bits, perplexity improves from 36.86 (uniform) to 11.73 (MSQ with Gaussian-assumed error), and further to 10.76 when using true quantization error. A similar trend holds at \~4 bits, where MSQ improves over the uniform baseline from 6.21 to 6.17 (Gaussian-assumed error), and further to 6.13 (true quantization error). Although applying IP consistently yields the best performance (6.84 at 3 bits, 5.92 at 4 bits), these results confirm that the bit allocation algorithm remains effective even without IP.
>
> **Experimental details**. In the no-IP setting, directly using Q-Palette’s per-tensor codebooks (trained on Gaussian-distributed weights) leads to severe performance degradation due to distribution mismatch. To mitigate this, we replaced Q-Palette’s per-tensor codebooks with per-column codebooks fitted to each column of the weight matrix via k-means clustering. We used a pool of NUQ quantizers ranging from 2 to 8 bits for MSQ. Bit allocation was computed by solving the ILP in Problem (3) of the paper, where the per-layer quantization error $\\text{err}(Q_q; W_l)$ was instantiated either as (a) precomputed distortion on Gaussian weights (“Gaussian-assumed error”), or (b) actual distortion measured from the retrained codebooks without IP (“true quantization error”).
>
> In conclusion, while incoherence processing provides the most favorable quantization performance, our bit allocation algorithm maintains strong performance even without it. We will include these ablation results and analysis in Appendix G of the camera-ready version.
>
> ---
>
> **W2. Theorem 3.1 seems suspicious to me. max(0, bitrate) implies that if the computed bitrate is negative, this linear layer should be pruned from the model since it gets 0 bits. Surely this isn't correct?**
>
> The derivation of Theorem 3.1 is mathematically correct under its stated assumptions. The theorem is derived by minimizing a surrogate objective based on the linearity theorem, assuming that optimal distortion quantizers exist for any non-negative bitwidth. Under this formulation, it is not mathematically incorrect for the optimal bitwidth of extremely low-sensitivity layers to become zero.
>
> Although this behavior may appear counterintuitive, it is not due to a flaw in the theorem, but rather reflects the limitations of the surrogate approximation. The surrogate is based on a second-order Taylor expansion of the perplexity loss and is accurate when quantization errors are small. However, at very low bitwidths, the quantization error becomes large, and the surrogate may incur significant approximation errors, leading to seemingly unintuitive solutions in extreme cases.
>
> To prevent such extreme solutions caused by surrogate inaccuracies, a practical refinement is to introduce a minimum bitwidth  threshold \$\eta\$, restricting the optimization to bitwidths \$\ge \eta\$, where the surrogate approximation expected to remain valid. This leads to a refined solution:
>
> $$
> b_l^* = \\max\\left\\{\\eta,\\frac{1}{2\\ln(2)}\\left(\\ln\\frac{a_l}{d_l^{\\mathrm{in}} d_l^{\\mathrm{out}}}\\right) + C\\right\\},
> $$
>
> with \$C\$ chosen to satisfy the overall memory constraint.
>
> In practice, we already adopt this principle by including only quantizers of 1.5 bits or higher in Q-Palette. Our solver never assigns zero-bitwidths, and all layers are quantized with strictly positive values. We will clarify this discussion and incorporate the refined version of the theorem in Appendix A of the camera-ready version.
>
> ---
>
> **W3. Do you have experiments that show that the bit allocation algorithm performs worse when used in the VQ or SQ case?**
>
> Yes, we provide such comparisons in Figure 3 of the paper. Figure 3 visualizes the WikiText2 perplexity achieved by our memory-constrained MSQ algorithm using different sets of quantizers under varying bitwidth constraints.
>
> * The **blue curve** corresponds to MSQ using only **TCQ quantizers at 2, 3, and 4 bits**.
> * The **cyan curve** corresponds to MSQ using only **VQ quantizers at 2, 3, and 4 bits**.
>
> As shown, the blue curve (`TCQ-2,3,4`) consistently outperforms the cyan curve (`VQ-2,3,4`) across all tested bitwidth constraints. This empirically supports our claim that the bit allocation algorithm is more effective when used with quantizers that better approximate the rate-distortion limit, such as TCQ. Compared to VQ and SQ, TCQ achieves lower distortion at the same bitwidth (see Figure 2), resulting in better perplexity-bitwidth tradeoffs.
>
> We will clarify this interpretation of Figure 3 in the camera-ready version to explicitly support the reviewer’s question.

---

> > ### Comment · Reviewer_hvcr · 2025-08-01
> >
> > > Do you have experiments that show that the bit allocation algorithm performs worse when used in the VQ or SQ case?
> >
> > What I meant by this question was whether the bit allocation algorithm does a worse job of allocating bits when used with a quantizer that is farther away from the rate-distortion limit. That is, is the gap between the optimal allocation and the solution found by the bit allocation algorithm larger for VQ and SQ than for TCQ due to VQ/SQ having worse rate distortion tradeoffs than TCQ? Figure 3 shows that VQ and SQ perform worse, which is expected since they have worse distortion than TCQ, but I don't think it answers this question.

---

> > > ### Author Response · Authors · 2025-08-02
> > >
> > > **Response to Additional Comment 1.**
> > >
> > > Thank you for the clarification. We now understand that your question focuses not only on the end-to-end performance of MSQ under different quantizer sets (*e.g.*, TCQ-2,3,4 vs. VQ-2,3,4), but specifically on the optimality gap, *i.e.*, how suboptimal the solution of MSQ is compared to the theoretical ideal derived in Theorem 3.1.
> > >
> > > Formally, let
> > > $\\{ b_l^\\ast \\}_{l=1}^L$
> > > denote the ideal bit allocation under ideal Gaussian quantizers, and
> > >
> > > $\\{Q_l^\ast\\}_{l=1}^L \in \\mathcal{Q}^L$ be the optimal quantizer assignment given a quantizer set $\mathcal{Q}$. Then, the optimality gap can be decomposed as:
> > >
> > >
> > > $$
> > > \underbrace{\sum_{l=1}^L a_l (\mathrm{err}(Q_l^\ast) -2^{-2b_l^\ast})}_{\text{Total gap}}
> > > $$
> > >
> > >
> > > $$
> > > = \underbrace{\sum_{l=1}^L a_l (\mathrm{err}(Q_l^\ast) - 2^{-2 \mathrm{bit}(Q_l^\ast)})}_{\text{Distortion gap}}
> > > $$
> > >
> > > $$
> > > +
> > > \underbrace{\sum_{l=1}^L a_l \left(2^{-2 \mathrm{bit}(Q_l^\ast)} - 2^{-2 b_l^\ast}\right)}_{\text{Bit allocation gap}}.
> > > $$
> > >
> > > This decomposition is discussed in Section B of the supplementary material. As you rightly point out, Figure 3 primarily reflects end-to-end performance and does not distinguish whether the performance gap is due to bit allocation or quantizer distortion.
> > >
> > > To address your question, we report the distortion gap and the bit allocation gap for various quantizer sets on LLaMA 3.1-8B under 2.5- and 3.25-bit constraints:
> > >
> > > *Table E. Optimality gap analysis for different quantizer sets on LLaMA 3.1-8B. TCQ-All includes all fractional TCQ bitwidths from 1.5 to 5.0 in Q-Palette.*
> > >
> > > | Quantizer pool        | Bitwidth | Distortion gap ↓ | Bit allocation gap ↓ | Total gap ↓ | Surrogate objective ↓ |
> > > | --------------------- | -------- | ---------------- | -------------------- | ----------- | --------------------- |
> > > | VQ-2,3,4              | 3.25     | 0.0586           | 0.0130               | 0.0716      | 0.1219                |
> > > | TCQ-2,3,4             | 3.25     | 0.0198           | 0.0129               | 0.0327      | 0.0830                |
> > > | TCQ-All               | 3.25     | 0.0145           | 0.0023               | 0.0168      | 0.0671                |
> > > | Ideal Gaussian quant. | 3.25     | 0                | 0                    | 0           | 0.0503                |
> > > | VQ-2,3,4              | 2.50     | 0.1282           | 0.0178               | 0.1460      | 0.2883                |
> > > | TCQ-2,3,4             | 2.50     | 0.0306           | 0.0178               | 0.0484      | 0.1907                |
> > > | TCQ-All               | 2.50     | 0.0260           | 0.0034               | 0.0294      | 0.1717                |
> > > | Ideal Gaussian quant. | 2.50     | 0                | 0                    | 0           | 0.1423                |
> > >
> > > These results lead to two key observations:
> > >
> > > 1. **Effect of quantizer quality.** VQ-2,3,4 and TCQ-2,3,4 exhibit similar bit allocation gaps, but TCQ-2,3,4 shows significantly smaller distortion gaps. This suggests that the performance gap between VQ-2,3,4 and TCQ-2,3,4 is primarily due to differences in quantizer quality (distortion), not bit allocation.
> > >
> > > 2. **Effect of broader bitwidth support.** Comparing TCQ-2,3,4 to TCQ-All reveals a substantial reduction in the bit allocation gap, indicating that the broader bitwidth support enables more accurate bit allocation and thus yields a closer approximation to the theoretical ideal.
> > >
> > > These analyses refine the interpretation of Figure 3 and will be included in Appendix B of the camera-ready version. We thank the reviewer for raising this point, which helped us better highlight the structure of the optimality gap and the practical implications of quantizer design choices.

---

> > ### Comment · Reviewer_hvcr · 2025-08-01
> >
> > >W2. Theorem 3.1 seems suspicious to me. max(0, bitrate) implies that if the computed bitrate is negative, this linear layer should be pruned from the model since it gets 0 bits. Surely this isn't correct?
> >
> > The extreme case I was thinking of was if an entire decoder layer got pruned, then the model would break in the middle. This paper would be strengthened by having an empirical analysis of allowing MSQ to assign 0 bits (note that this doesn't require a new quantizer, since 0 bits is just pruning the layer) and seeing if it ever does this.

---

> > > ### Author Response · Authors · 2025-08-02
> > >
> > > **Response to Additional Comment 2.**
> > >
> > > Thank you for raising this interesting point regarding the interpretation of Theorem 3.1. We agree that the expression `max(0, ...)` could imply the assignment of 0 bits to a layer, which may be interpreted as layer pruning. While the theorem is mathematically valid under the stated assumptions, in practice, pruning intermediate layers can break the model in the middle and is therefore undesirable.
> > >
> > > As you suggested, we assessed whether our method ever assigns 0 bits in practice, we conducted an empirical analysis by explicitly including a 0-bit "quantizer" (layer pruning) as an additional option in the quantizer set. We then solved the MSQ optimization problem under various bitwidth constraints on LLaMA 3.1-8B. The results are summarized below:
> > >
> > > *Table F. Effect of including a 0-bit quantizer in MSQ optimization on LLaMA 3.1-8B. “# Pruned layers” denotes the number of layers assigned 0 bits by the optimizer.*
> > >
> > > | Quantizer pool            | Bitwidth | # Pruned layers  | Surrogate objective ↓ | Wiki2 perplexity ↓ |
> > > | ------------------------- | --------- | ---------------- | --------------------- | ------------------ |
> > > | TCQ-All (1.5–5.0)         | 4.00      | 0                | 0.0294                | 5.81               |
> > > | TCQ-All + pruning (0.0)   | 4.00      | 0                | 0.0294                | 5.81               |
> > > | Ideal Gaussian quantizers | 4.00      | 0                | 0.0178                | –                  |
> > > | TCQ-All                   | 3.25      | 0                | 0.0671                | 6.10               |
> > > | TCQ-All + pruning    | 3.25      | 0                | 0.0671                | 6.10               |
> > > | Ideal Gaussian quantizers | 3.25      | 0                | 0.0503                | –                  |
> > > | TCQ-All                   | 3.00      | 0                | 0.0908                | 6.28               |
> > > | TCQ-All + pruning    | 3.00      | 0                | 0.0908                | 6.28               |
> > > | Ideal Gaussian quantizers | 3.00      | 0                | 0.0712                | –                  |
> > > | TCQ-All                   | 2.00      | 0                | 0.3341                | 9.20               |
> > > | TCQ-All + pruning    | 2.00      | 1 (0-th q\_proj) | 0.3336                | 615.8              |
> > > | Ideal Gaussian quantizers | 2.00      | 1 (0-th q\_proj) | 0.2847                | –                  |
> > >
> > > As shown, for bitwidths 3.0, 3.25, and 4.25 (settings of Table 3 in the paper), no layers were assigned 0 bits, even when pruning was allowed. This aligns with the ideal Gaussian solution, which also avoids zero-bit assignments under bitwidths 3.0, 3.25, and 4.25.
> > >
> > >
> > > However, under the more aggressive 2-bit constraint, the solver pruned a single layer (the 0-th q\_proj layer), yielding a marginally better surrogate objective. Yet, this choice caused a catastrophic degradation in perplexity (615.8).
> > > This behavior highlights a key limitation of the surrogate objective: it can significantly underestimate the true degradation caused by pruning. As we mentioned in our previous response, this issue can be mitigated in practice by enforcing a minimum bitwidth threshold (*e.g.*, $\eta = 1.5$) during optimization.

---

> > ### Comment · Reviewer_hvcr · 2025-08-01
> >
> > >In the no-IP setting, directly using Q-Palette’s per-tensor codebooks (trained on Gaussian-distributed weights) leads to severe performance degradation due to distribution mismatch.
> >
> > If you have time, could you rerun this experiment with a fixed quantizer (eg gaussian) and group scaling? This is more realistic than fitting a codebook to each group.

---

> > > ### Author Response · Authors · 2025-08-02
> > >
> > > **Response to Additional Comment 3.**
> > >
> > > Thank you for the thoughtful suggestion. As requested, we reran the experiment using per-group quantization (group size = 64) with a fixed codebook trained on Gaussian-distributed weights, as used in Q-Palette. All results are obtained without IP. The results are summarized in Table G below:
> > >
> > > *Table G. Per-group quantization results on LLaMA 3.1-8B (group size = 64). For MSQ, we used a pool of per-group NUQ quantizers ranging from 2 to 8 bits with group size 64. All results are reported without IP.*
> > >
> > > | Method       | Bit allocation strategy                   | Bitwidth | Wiki2 perplexity ↓ |
> > > | ------------ | ----------------------------------------- | -------- | ------------ |
> > > | Ours-NUQ-3   | Uniform bitwidth (3-bit, group size = 64) | 3.25     | 8063.91      |
> > > | Ours-MSQ-Mem | Gaussian-assumed error                    | 3.25     | 7.34         |
> > > | Ours-MSQ-Mem | True quantization error                   | 3.25     | 7.33         |
> > > | Ours-NUQ-4   | Uniform bitwidth (4-bit, group size = 64) | 4.25     | 6.10         |
> > > | Ours-MSQ-Mem | Gaussian-assumed error                    | 4.25     | 5.89         |
> > > | Ours-MSQ-Mem | True quantization error                   | 4.25     | 5.90         |
> > >
> > > As shown in Table G, even when using fixed Gaussian-trained codebooks with per-group scaling, our MSQ method outperforms uniform baselines. We will include these results in Appendix G of the camera-ready version.

---

> ### Comment · Reviewer_hvcr · 2025-08-03
>
> Thanks for your follow up experiments. I think they confirm my initial thoughts about the paper, which is that the bit allocation algorithm and better tcq kernels are somewhat orthgonal to each other.  The bit allocation theorem could also be improved due to the possibility of just breaking the model in half (which apparently happens empirically without a bit floor). However, I think both components of the paper are useful and practical contributions that are not just incremental. I think this paper should be accepted, and if we were on the old scale I would give it a weak accept, which seems to be a 4.5 on this scale. I will keep my score at 4 for now since IMO a 5 would be a 7-8 on the old scale, and I don't think this paper is at the level of 7-8.

---

### Official Review · Reviewer_LLDE · 2025-06-29

**Clarity:** 3
**Significance:** 3
**Originality:** 3
**Rating:** 4
**Confidence:** 4

**Summary:**

This paper presents a set of fractional-bit quantizers aimed at accelerating weight-only PTQ for LLMs. The proposed CUDA kernels, together with a mixed-scheme quantization framework, demonstrate competitive performance in decoding speed.

**Questions:**

1. What is the effectiveness of Q-Palette and MSQ on larger LLMs (e.g., LLaMA3-70B) and on other architectures such as the Qwen series?
2. Could Q-Palette be extended to support or improve weight-activation quantization methods, particularly rotation-based techniques?

**Ethical Concerns:**

["NO or VERY MINOR ethics concerns only"]

**Limitations:**

yes

**Paper Formatting Concerns:**

No.

**Quality:**

3

**Strengths And Weaknesses:**

**Strengths**

1. The custom CUDA kernel implementation is technically solid and shows real-world speedups on NVIDIA GPUs.
2. The quantizers introduced in Q-Palette achieve significant decoding speed improvements compared to existing baselines such as NF and QTIP.

**Weaknesses**

1. The experimental evaluation is primarily limited to LLaMA series models and small-scale models. Broader coverage would strengthen the conclusions.
2. Some relevant rotation-based quantization methods, such as DuQuant [1] and OstQuant [2], are not discussed in the Related Work section. Please consider citing and briefly discussing these in Section 6.
3. The solution to fusion-based mixed-scheme quantization in Section 4.2 is not clearly explained. Additional details are needed to understand how the solution is derived and implemented.


[1]. DuQuant: Distributing Outliers via Dual Transformation Makes Stronger Quantized LLMs. NeurIPS 2024.

[2]. OstQuant: Refining Large Language Model Quantization with Orthogonal and Scaling Transformations for Better Distribution Fitting. ICLR 2025.

---

> ### Author Rebuttal · Authors · 2025-07-30
>
> We would like to express our gratitude for your encouraging comments regarding our “technically solid” CUDA kernel implementation and the “significant decoding speed improvements” achieved by Q-Palette.
>
> ---
> **W1 & Q1. What is the effectiveness of Q-Palette and MSQ on larger LLMs (e.g., LLaMA3-70B) and on other architectures such as the Qwen series?**
>
> Thank you for raising this important point. We agree that evaluating our framework on both larger-scale and non-LLaMA architectures is crucial to demonstrating generality.
>
> To this end, we applied our MSQ method with Q-Palette to LLaMA 3.1-70B (large-scale) and Qwen 2.5-7B (non-LLaMA), comparing against HIGGS-MSQ [3] under various bitwidth constraints.
>
>
> *Table A. Mixed-scheme quantization results on Qwen 2.5-7B. The FP16 perplexity is 6.13.*
>
> | Bitwidth | Wiki2 perplexity of Ours-MSQ-Mem (↓) | Wiki2 perplexity of HIGGS-MSQ (↓) |
> | -------- | ------------------------------------ | --------------------------------- |
> | 3.25     | **6.41**                             | 6.60                              |
> | 3.50      | 6.33                                 | -                                 |
> | 3.75     | 6.27                                 | -                                 |
> | 4.00      | **6.24**                             | 6.33                              |
> | 4.25     | **6.21**                             | 6.28                              |
>
> *Table B. Mixed-scheme quantization results on LLaMA 3.1-70B. The FP16 perplexity is 2.54. To accommodate the broader sensitivity range in LLaMA 3.1-70B, we extended the quantizer set to include higher-bitwidth options (NUQ 7/8 bits and VQ 5.5/6 bits), in addition to the TCQ quantizers.*
>
> | Bitwidth | Wiki2 perplexity of Ours-MSQ-Mem (↓) | Wiki2 perplexity of HIGGS-MSQ (↓) |
> | -------- | ------------------------------------ | --------------------------------- |
> | 3.25     | **3.28**                             | 3.68                              |
> | 3.40      | 3.06                                 | -                                 |
> | 3.63    | 2.94                                 | -                                 |
> | 4.00      | **2.77**                             | 3.13                              |
> | 4.25     | **2.70**                             | 2.96                              |
>
> As shown in Tables A and B, our method consistently outperforms HIGGS-MSQ under the same bitwidth constraints (3.25, 4.00, 4.25) on both models. Additionally, our method achieves comparable or better perplexity at lower bitwidths compared to HIGGS-MSQ. For Qwen 2.5-7B, our 3.50-bit model matches the performance of HIGGS-MSQ at 4.00 bits (6.33), and our 3.75-bit result slightly improves upon the HIGGS-MSQ result at 4.25 bits (6.27 vs. 6.28), yielding up to 12.5% memory savings. A similar trend is observed on LLaMA 3.1-70B, where our 3.40-bit and 3.63-bit results slightly outperform HIGGS-MSQ at 4.00 and 4.25 bits, respectively, resulting in up to 15% memory savings at better perplexity.
>
> These results demonstrate the broad applicability of our framework. We will include these additional results in Appendix G of the camera-ready version.
>
>
>
> ---
>
> **Q2. Could Q-Palette be extended to support or improve weight-activation quantization methods, particularly rotation-based techniques?**
>
> In this work, we focus on weight-only PTQ, which is particularly effective in memory-bound inference settings with small batch sizes, such as on laptops or mobile devices where memory bandwidth, rather than compute, is the primary bottleneck.
>
> However, on hardware accelerators like the Qualcomm Hexagon NPU in Snapdragon 8 Gen 3, which natively support only integer (*e.g.*, INT8) GEMM, activation quantization is essential for leveraging their full performance. Thus, extending Q-Palette to support weight-activation quantization is a promising direction for broader deployment.
>
> One potential approach is a two-stage scheme: (1) first quantize weights to INT8 using uniform W8A8 quantizers for hardware compatibility, and (2) then apply a secondary compression step that further quantizes the INT8 weights into x-bit representations using a variant of Q-Palette quantizers whose codebooks are constrained to the INT8 grid. During inference, the compressed weights are dequantized back to INT8 and then processed using integer GEMM with INT8 quantized activations, enabling compatibility with INT8-only hardware while reducing memory usage.
>
> We consider this an important direction for future work and will discuss it in Appendix H of the camera-ready version.
>
> ---
>
> **W2. Some relevant rotation-based quantization methods, such as DuQuant [1] and OstQuant [2], are not discussed in the Related Work section. Please consider citing and briefly discussing these in Section 6.**
>
> Thank you for pointing this out. We will revise Section 6 to cite and briefly discuss recent rotation-based quantization methods including DuQuant [1] (which applies rotation and permutation), OstQuant [2] (which uses learned scaling and rotation matrices), and FlatQuant [4] (which adopts learned affine transformations). These methods primarily target weight-activation quantization, whereas our work focuses on weight-only PTQ which is especially suited for memory-bound, small-batch inference. We will clarify this distinction and update the Related Work section accordingly in the camera-ready version.
>
> ---
>
>
>
>
>
>
> **W3. The solution to fusion-based mixed-scheme quantization in Section 4.2 is not clearly explained. Additional details are needed to understand how the solution is derived and implemented.**
>
> Thank you for your question regarding the fusion-aware MSQ formulation in Section 4.2. We are happy to provide further clarification.
>
> We begin by briefly reviewing operator fusion. Within each Transformer block, certain linear layers, such as {query, key, value} or {up, gate}, share the same input and can be fused into a single matrix multiplication. For example, instead of separately computing $ \\mathbf{x}W_q$, $ \\mathbf{x}W_k$, and $ \\mathbf{x}W_v$ for an input $\\mathbf{x}$, we can concatenate the weight matrices and compute $\\mathbf{x}(W_q \oplus W_k \oplus W_v) $, followed by splitting the output. This common optimization is often used to improve inference speed by reducing the number of kernel launches and memory accesses. A visualization of fused layers is provided on the right side of Figure 1 in our paper.
>
>
> In quantized models, however, fusion is only valid if the involved layers share both the input and **the same quantization configuration**. Fusion-aware MSQ is designed to jointly determine:
>
> 1. how to group layers for fusion, and
> 2. which quantizer to assign to each fused group.
>
> Whereas standard MSQ (Section 4.1) introduces one binary variable per (layer, quantizer) pair, fusion-aware MSQ defines one binary variable per (fusible layer group, quantizer) pair. For each Transformer block $b$, we define the set of fusible layer groups:
>
> $$
> \mathcal{G}_b=\\{\\{q_b\\}, \\{k_b\\}, \\{v_b\\}, \\{q_b,k_b\\}, \\{q_b,v_b\\}, \\{k_b,v_b\\}, \\{q_b,k_b,v_b\\}, \\{o_b\\}, \\{u_b\\}, \\{g_b\\}, \\{u_b,g_b\\}, \\{d_b\\}\\}.
> $$
>
> and the full set is $\mathcal{G} = \\bigcup_{b=1}^B \mathcal{G}_b$, where $B$ is the number of Transformer blocks.
>
> Each binary variable $P_{gq} \in \\{0,1\\}$ indicates whether the layer group $g$ is quantized using quantizer $q$. These variables jointly encode both the fusion and quantization decisions.
>
> To ensure valid solutions, the problem imposes two constraints:
>
> 1. **Exclusive assignment**: each layer must appear in exactly one activated (group, quantizer) pair, *i.e.*, among all groups $g$ that contain layer $l$, only one corresponding $P_{gq}$ may be 1:
>
> $$
> \sum_{g \in \mathcal{G}: l \in g} \sum_{q=1}^{|\mathcal{Q}|} P_{gq} = 1, \quad \forall l \in \bigcup_{b=1}^B \\{q_b, k_b, v_b, o_b, u_b, g_b, d_b\\}.
> $$
>
> 2. **Resource constraint**: the total profiled cost (*e.g.*, latency or memory) of all activated groups must not exceed the resource budget $C$:
>
> $$
> \sum_{g \in \mathcal{G}} \sum_{q=1}^{|\mathcal{Q}|} P_{gq} \cdot c_{gq} \le C,
> $$
>
> where $c_{gq}$ denotes the profiled cost (*e.g.*, latency) of executing the fused layer group $g$ quantized by $q$.
>
> The objective is to minimize total estimated performance loss across all groups:
>
> $$
> \sum_{g \in \mathcal{G}} \sum_{q \in \mathcal{Q}} P_{gq} \cdot \sum_{l \in g} \ell_{lq},
> $$
>
> where $\ell_{lq}$ is the estimated loss from quantizing layer $l$ with quantizer $q$. The resulting optimization problem corresponds to Problem (4) in our paper.
>
> Since both the objective and constraints are linear in $P_{gq}$, the problem can be formulated as an integer linear program (ILP). We solve this ILP problem using the SCIP solver in OR-Tools and our implementation typically completes within seconds to minutes across all experiments. For implementation details, please refer to the supplementary material (`solve_optimal_quantizer` function in `codes/solve_lat_const.py`).
>
> We believe that fusion-aware MSQ is one of the key contributions of our work. To the best of our knowledge, it is the first quantization framework to jointly optimize **layer fusion and quantizer assignment** under deployment constraints within a unified, practical ILP formulation. We will revise the camera-ready version to make this contribution more prominent and clearly presented.
>
>
> ---
> References
>
> [1]. DuQuant: Distributing Outliers via Dual Transformation Makes Stronger Quantized LLMs. NeurIPS 2024.
>
> [2]. OstQuant: Refining Large Language Model Quantization with Orthogonal and Scaling Transformations for Better Distribution Fitting. ICLR 2025.
>
> [3]. Pushing the Limits of Large Language Model Quantization via the Linearity Theorem. NAACL 2025.
>
> [4]. FlatQuant: Flatness Matters for LLM Quantization. ICML 2025.

---

> > ### Comment · Reviewer_LLDE · 2025-08-05
> >
> > Thank you for the additional results and clarifications. I maintain my score in support of this paper. Please incorporate these discussions into the next version of the manuscript.

---

> > > ### Author Response · Authors · 2025-08-07
> > >
> > > Thank you for your careful review, valuable feedback, and supportive assessment of our work. We will incorporate the additional results and clarifications from the discussion into the final manuscript.

---

### Official Review · Reviewer_skQe · 2025-07-03

**Clarity:** 3
**Significance:** 2
**Originality:** 1
**Rating:** 3
**Confidence:** 3

**Summary:**

This paper proposes a multi-scheme quantization framework for post-training quantization of neural networks, including large language models (LLMs). Specifically, it formulates the problem of choosing the most suitable quantizer for each layer—selected from a suite of quantizers with varying precision levels—as an optimization problem that can be numerically solved to balance accuracy and model size. The authors further extend their approach to account for layer fusion, noting that the fusion-aware optimization problem is NP-hard but can still be addressed numerically in practice.

**Questions:**

- Can the computation of the layer sensitivities $\alpha_l$ be explained in greater detail, particularly in the context of data-free settings?
- Do the experimental results include the fusion-aware framework? Does including/not including it make a significant difference in the results?
- Does the linearization of the model loss hold for non-standard architectures such as those with many skip layer connections?

**Ethical Concerns:**

["NO or VERY MINOR ethics concerns only"]

**Final Justification:**

I thank the authors for their detailed response. Despite the noteworthy practical value demonstrated, as stated in my final response to the authors, I will keep my original score of 3 because of my concerns with the novelty.

**Limitations:**

yes

**Quality:**

2

**Strengths And Weaknesses:**

The paper’s main strengths are its practical contributions: the authors provide efficient CUDA implementations of the quantizers used in their framework, making the method readily usable for practitioners interested in deploying their solution. Moreover, the experimental results show some improvement over both uniform bitwidth baselines and a previous work that allocated bit-widths to layers non-uniformly.

The main weakness of the paper lies in its limited novelty. The quantizer selection through optimization closely parallels the principles established in Malinovskii et al. ([30]) for bitwidth allocation, and the approach appears to build directly on that prior framework. Additionally, the quantizers included in the QPalette suite are classical, well-known algorithms, and the work does not introduce new quantization techniques. As a result, while the practical contributions are clear and valuable for deployment, the paper’s research novelty may not be sufficient to justify acceptance.

---

> ### Author Rebuttal · Authors · 2025-07-31
>
> We thank the reviewer for recognizing the practical value of our work, including our efficient CUDA kernel implementations and improvements over uniform and non-uniform bitwidth baselines.
>
> ---
>
> **W. Regarding novelty and contributions beyond prior work**:
>
> While Q-Palette builds on established quantizer families (NUQ, VQ, TCQ), we emphasize that supporting efficient kernel execution for fine-grained fractional-bit variants and larger batch sizes required substantial engineering effort, including the precise mapping of quantized weights to warp-level `mma` instruction fragments for each quantizer configuration. As the reviewer acknowledged, these kernels make Q-Palette “readily usable for practitioners” and contribute significantly to real-world deployability.
>
> Beyond these engineering aspects, our work introduces several novel contributions that distinguish it from prior work, including HIGGS \[30].
>
> **First, we propose fusion-aware MSQ, addressing the reviewer’s concern that our optimization framework closely follows HIGGS \[30].**
>
> As stated in the paper, our standard MSQ formulation in Section 4.1 adopts a well-established mixed-precision quantization framework based on the linear surrogate loss introduced in prior works \[13, 6, 30]. We use this formulation to evaluate the effectiveness of our quantizer set (Q-Palette) under memory constraints, but we do not claim it as our contribution.
>
> In contrast, under latency-constrained scenarios, we introduce fusion-aware MSQ (Section 4.2), which extends the standard MSQ formulation by jointly optimizing quantizer assignment and operator fusion decisions. Unlike standard MSQ, which considers only the cost and loss associated with each individual layer, fusion-aware MSQ incorporates the potential cost reduction from operator fusion directly into the optimization process, enabling additional opportunities for performance improvement. This leads to substantial improvements in the accuracy-latency trade-off in practical deployment scenarios. The empirical results demonstrating the effectiveness of fusion-aware MSQ are provided in our response to Q2.
>
>
> **Second, we introduce half-TCQ, a simple yet novel variant that enables finer control over bitwidths.**
>
> While Q-Palette builds upon established quantizer families, it also includes new extensions that go beyond prior work. In particular, half-TCQ splits a weight matrix and applies different TCQ bitwidths to each half (e.g., 2.5 and 3.0 bits), achieving a 2.75-bit representation. This design enables finer-grained bitwidth control beyond the 0.5-bit intervals. To support efficient deployment, we implement CUDA kernels that perform dequantization and matrix multiplication for a half-TCQ layer in a single kernel call. To our knowledge, half-TCQ is a novel quantization technique that has not appeared in prior work and directly addresses the reviewer’s concern regarding the lack of new quantization techniques.
>
>
> **Third, our theoretical analysis in Section 3.1 provides a principled motivation for the design of Q-Palette.**
>
> Theorem 3.1 and the analysis in Appendix B show that to improve MSQ performance, it is important to use practical quantizers that achieve distortion close to the theoretical optimum for a given bitwidth, and are available at fine-grained fractional bitwidths. This insight motivates the design of Q-Palette and offers guidance for future quantizer development tailored to optimal bit allocation.
>
> We plan to revise the introduction in the camera-ready version to more explicitly highlight these contributions.
>
> ---
>
> **Q1. Can the computation of the layer sensitivities be explained in greater detail, particularly in the context of data-free settings?**
>
> Thank you for the question. The procedure for computing the sensitivity coefficients $a_l$ in data-free settings is described in **Appendix E.1**. We briefly summarize it here for clarity.
>
> In data-free scenarios, we estimate the surrogate loss as
>
> $$
> \ell_{lq} = a_l \cdot \mathrm{err}(Q_q; W_l),
> $$
>
> where $\mathrm{err}(Q_q; W_l)$ is approximated using the precomputed distortion of quantizing standard Gaussian matrices. To compute the sensitivity coefficient $a_l$, we adopt the procedure introduced in the HIGGS \[30]. Specifically, we:
>
> * Generate 128K random tokens from the LLM. We used `top_k=50`, `top_p=0.98` with `temperature=1.0` for generation.
> * Inject scaled Gaussian noise into each weight matrix $W_l$ at 16 different norm levels
>   $n_{li} = \frac{\sqrt{i}}{16} \\|W_l\\|_2$ for $i = 1, \dots, 16$.
> * Measure how much the model’s KL-divergence loss increases on randomly sampled 128K tokens when Gaussian noise of each level is injected into layer $l$.
>
> This increase in loss is approximately linear in $n_{li}^2$, and we estimate $a_l$ by linear regression on ($n_{li}^2$, increase in loss) data. This process requires $16 \times L$ forward evaluations of the loss function but can be done in embarassingly parallel manner. Once computed, the sensitivity coefficients can be reused across all data-free MSQ scenarios with no additional cost.
>
> If the explanation in Appendix E.1 was unclear, we hope this summary helps clarify the practical steps and rationale.
>
>
> ---
>
> **Q2. Do the experimental results include the fusion-aware framework? Does including/not including it make a significant difference in the results?**
>
> Thank you for your question regarding the fusion-aware MSQ (Section 4.2). We demonstrate the impact of fusion-aware MSQ through experiments comparing it to standard MSQ.
>
> First of all, Figure 1 in our paper shows the qualitative results on effectiveness of fusion-aware MSQ. In Figure 1(c), we show the quantizer allocation using the standard MSQ without considering fusion, while Figure 1(d) presents the allocation achieved with fusion-aware MSQ. As seen in the figure, the fusion-aware MSQ outperforms the standard MSQ in both perplexity and speedup. Specifically, **fusion-aware MSQ** achieves a perplexity of **6.39** and a speedup of **3.17x** which outperforms  the standard MSQ, which achieves a perplexity of 6.46 and a speedup of 2.99x,  in both perplexity and speedup. This demonstrates that fusion-aware MSQ leads to better accuracy while maintaining a higher speedup.
>
> Additional quantitative results are presented in Table 7 of Appendix G in supplementary material, where we compare the perplexity of fusion-aware MSQ against standard MSQ on LLaMA 3.1-8B across various throughput levels. For example, on an RTX 4090, with batch size 1, standard MSQ achieves 20.33 perplexity at 223 tokens/sec, while our **fusion-aware MSQ** achieves a significantly reduced perplexity of **7.79** at **224** tokens/sec.
>
> These results confirm that the fusion-aware framework brings substantial improvements over standard MSQ. We will further emphasize these findings in our camera-ready version to ensure the impact of fusion-aware MSQ is clearly communicated.
>
>
> ---
>
> **Q3. Does the linearization of the model loss hold for non-standard architectures such as those with many skip layer connections?**
>
> Thank you for the thoughtful question. We would like to clarify that standard transformer architectures, including all models evaluated in our paper (LLaMA 2-7B, 13B, LLaMA 3.2-1B, 3B, and 3.1-8B), already include two skip connections per block: one around the self-attention module and another around the MLP. As such, our results provide empirical evidence that the linearization-based surrogate objective works effectively in architectures with multiple skip connections.
>
> We do not analyze the surrogate’s behavior in architectures with significantly different connectivity patterns, such as DenseNet  for computer-vision tasks. Exploring such cases could be worthwhile in broader contexts, but this falls outside the scope of our study, which focuses on quantizing pretrained Transformer-based LLMs.

---

> > ### Comment · Reviewer_skQe · 2025-08-05
> >
> > Thank you for your detailed response and for addressing my questions. I appreciate the novel aspects of your approach that you have highlighted. Nonetheless, I continue to feel that the overall novelty is somewhat limited, and will therefore be maintaining my original score. I would, however, not be opposed to acceptance should the other reviewers be in agreement.

---

> > > ### Author Response · Authors · 2025-08-07
> > >
> > > Thank you for your thoughtful review and constructive comments. We appreciate your recognition of our practical contributions and will incorporate the discussion points in the camera-ready version.

---

### Note · Authors · 2025-08-12

We thank the AC and reviewers for their time and constructive feedback. Below we summarize the main concerns raised, how we addressed them, and where details can be found in the rebuttal threads.

---

**Novelty and contribution**

*Reviewers skQe and 4xNx* noted that our MSQ “closely parallels” HIGGS and that our quantizers are not new. We clarified that we do not claim the standard MSQ method as our contribution, as it is a well-established approach used in prior works such as HIGGS. Our novel contribution is the fusion-aware MSQ, which jointly optimizes quantizer allocation and operator fusion decisions. We also clarified how our TCQ differs from prior TCQ **(see [skQe W], [4xNx W1, W2, Q1])**.

**Details on fusion-aware MSQ**

*Reviewers LLDE and skQe* requested a detailed explanation of fusion-aware MSQ, and evidence of its effectiveness. We clarified its motivation, explained variables, objective, constraints, and how the ILP is solved. We also highlighted performance gains of fusion-aware MSQ over standard MSQ provided in Table 7 of Appendix G **(see [LLDE W3], [skQe Q2])**.

**Wide applicability**

*Reviewers LLDE and hvcr* raised concerns regarding the effectiveness of our method on other models and in settings without incoherence processing (IP). We addressed this by adding results on LLaMA 3.1-70B and Qwen 2.5-7B, showing our method still outperforms HIGGS-MSQ. We also performed ablations without IP, confirming MSQ still outperforms uniform baselines **(see [LLDE W1, Q1], [hvcr W1, AC3])**.

**Optimality gap and edge cases in bit allocation**

*Reviewer hvcr* raised concerns about the sources of the optimality gap among different quantizer sets, and possible zero-bit assignments in Theorem 3.1. Through constructive discussion, we clarified these points and reached a clear understanding with the reviewer **(see [hvcr W2, W3, AC1, AC2] for details)**.

---

*Three reviewers LLDE, hvcr, and 4xNx* expressed satisfaction with our responses and maintained positive scores. *LLDE* wrote, *“I maintain my score in support of this paper.”* *hvcr* noted, *“I think this paper should be accepted …”* *4xNx* stated, *“I’m generally satisfied with the overall contribution …”* The remaining *reviewer skQe*, remarked, *“I would, however, not be opposed to acceptance should the other reviewers be in agreement.”* We believe all substantive concerns have been addressed and respectfully ask the AC take them into consideration in the final decision.

---

### Decision · Program_Chairs · 2025-09-17

**Decision:**

Accept (poster)

**Comment:**

The paper considers the problem of post-training quantization of LLMs. It tries to identify the optimal bit allocation, and quantization schemes to use at each layer of an LLM for achieving optimal latency and quality tradeoffs. The paper poses this as a search problem over a discrete space and relies on integer linear program solvers to identify a good solution. The particular search space considered is as follows
- **quantization schemes**: non uniform scalar quantization implemented using k-means clustering, 2D vector quantization,  and Trellis-coded quantization (TCQ)
- **bits**: apart from the usual  integer bit quantization schemes, the paper also considers fractional bit quantizers.  For instance, the authors propose to different number of bits for quantization first half of the weights in a layer compared to the second half. This effectively mimics fractional bit quantization.

While this framework itself is not novel, the paper introduces a new dimension to the search space by introducing layer fusion, where linear layers with same input can be fused together for reducing memory access. The other main contribution of the is to provide efficient CUDA, Tensor core kernels for the above quantization schemes.

The empirical results in the paper look strong. Compared to many baselines, the proposed technique achieves better quality-latency tradeoffs. The being said, there were a number of concerns raised by reviewers
- the main contribution of the work seems to be in the design of kernels for supporting various quantization schemes at various bitwidths. Unfortunately the paper doesn't provide many details on the kernel implementation. Without these details, it is hard to understand where the gains are coming from. For instance, one of the reviewers wondered if the speedups are simply coming from  reducing the number of online Hadamard transforms from 14 to 4.
- the paper never discusses the computational cost of the technique. Given that it requires running various PTQ algorithms at every layer, it can easily be an order of magnitude slower than traditional PTQ techniques. It is not clear how well the technique scales to larger models.

Despite these limitations, I believe the paper could be interesting to practitioners. So I recommend accepting it. But I encourage the authors to take this feedback into account to improve the camera-ready version of the paper.